# Non-coding RNA *LEVER* sequestration of PRC2 can mediate long range gene regulation

Wei Wen Teo [1,9], Xinang Cao [1,2,9], Chan-Shuo Wu[1], Hong Kee Tan[1,3], Qiling Zhou[1], Chong Gao[4], Kim Vanuytsel [5,6], Sara S. Kumar[5,6], George J. Murphy[5,6], Henry Yang [1], Li Chai [4✉] & Daniel G. Tenen [1,7,8✉]

Polycomb Repressive Complex 2 (PRC2) is an epigenetic regulator required for gene silencing during development. Although PRC2 is a well-established RNA-binding complex, the biological function of PRC2-RNA interaction has been controversial. Here, we study the gene-regulatory role of the inhibitory PRC2-RNA interactions. We report a nuclear long non-coding RNA, *LEVER*, which mapped 236 kb upstream of the *β-globin* cluster as confirmed by Nanopore sequencing. *LEVER* RNA interacts with PRC2 in its nascent form, and this prevents the accumulation of the H3K27 repressive histone marks within *LEVER* locus. Interestingly, the accessible *LEVER* chromatin, in turn, suppresses the chromatin interactions between the *ε-globin* locus and *β-globin* locus control region (LCR), resulting in a repressive effect on *ε-globin* gene expression. Our findings validate that the nascent RNA-PRC2 interaction inhibits local PRC2 function in situ. More importantly, we demonstrate that such a local process can in turn regulate the expression of neighboring genes.

[1] Cancer Science Institute of Singapore, National University of Singapore, Singapore, Singapore. [2] Department of Medicine, Yong Loo Lin School of Medicine, National University of Singapore, Singapore, Singapore. [3] National University of Singapore, Graduate School for Integrative Sciences and Engineering, Singapore, Singapore. [4] Department of Pathology, Brigham and Women's Hospital, Boston, MA, USA. [5] Section of Hematology and Medical Oncology, School of Medicine, Boston University, Boston, MA, USA. [6] Center for Regenerative Medicine, Boston University and Boston Medical Center, Boston, MA, USA. [7] Harvard Stem Cell Institute, Harvard Medical School, Boston, MA, USA. [8] Harvard Initiative for RNA Medicine, Harvard Medical School, Boston, MA, USA. [9] These authors contributed equally: Wei Wen Teo, Xinang Cao. ✉email: lchai@bwh.harvard.edu; daniel.tenen@nus.edu.sg

Polycomb Repressive Complex 2 (PRC2) is an epigenetic regulator that plays essential roles in cellular differentiation and early embryonic development. These biological functions are achieved mainly through establishing and maintaining the global methylation landscape of lysine 27 on histone H3. In particular, the final H3K27 tri-methylated product (H3K27me3) generally marks transcriptionally inactive regions and is considered as the major epigenetic output through which PRC2 supervises the repressive genome[1].

However, in mammals, how PRC2 selectively represses target genes remains elusive. Many long non-coding RNAs (lncRNAs), such as XIST[2] and HOTAIR[3,4], were proposed to serve as cofactors to direct PRC2 to its specific targets in the mammalian genome. Importantly, this model highlights an axis of a specific lncRNA-PRC2-specific target gene loci.

This lncRNA-directed PRC2-mediated gene repression model has been questioned by several follow-up studies on PRC2-RNA interaction[5–18]. First, the promiscuous RNA-binding pattern of PRC2 in vitro and in cells challenged the selectivity of PRC2 for its specific lncRNA partners[5,6,12]. Second, RNA can inhibit PRC2 enzymatic activity[7,8,10] by competing with histone for PRC2-binding in vitro[14] and in cells[12]. Furthermore, extensive interaction between PRC2 and nascent RNAs was observed in mouse embryonic stem cells (ESCs) at transcriptionally active gene loci[6,12], which brings into question the transcription repressive effect by local PRC2-RNA interaction at these loci. In line with these findings, PRC2 occupancy and H3K27me3 levels at previously active gene loci were increased upon global RNA degradation[12] or transcription blockage[9].

While an increasing amount of evidence suggested that PRC2 interacts with nascent RNAs and such interaction inhibits PRC2 function, the consequential effects of such PRC2-inhibitory interaction on gene expression have been mostly proposed as 'gene autonomous'. The nascent RNAs sequester and inhibit local PRC2, thereby protect their own gene loci from being accidentally repressed[5,6,12,18]. Meanwhile, these nascent RNAs also locally reserve PRC2, preparing the complex to occupy and function quickly once chromatin repressive signals are detected[5,6,18]. These two functions theoretically maintain the current cellular state while renders adaptiveness to environmental changes, and phenotypically this has been supported by one recent study showing that global disruption of PRC2-RNA interaction led to differentiation defects in induced pluripotent stem cells (iPSC)[17]. Notably, while this RNA inhibiting PRC2 model is well supported mechanistically, the histone modification was the major readout in most studies and its direct effect on autonomous gene transcription was less evidenced due to technical limitations caused by the autonomous regulation.

Here, we report a non-autonomous gene-regulatory role of the inhibitory PRC2-RNA interaction. We identified a PRC2-interacting non-coding RNA (termed LEVER, short for the regulation of ε-globin through inactivating EZH2 by an upstream non-coding RNA), and we further characterized LEVER by direct-RNA Nanopore sequencing. LEVER RNA represses the expression of the neighboring ε-globin gene. Mechanistically, LEVER RNA inhibits PRC2 chromatin occupancy and thereby regulates H3K27me2/me3 levels at the LEVER locus. The epigenetic status at the LEVER locus in turn influences chromatin interactions involving the β-globin Locus Control Region (LCR) that further regulates ε-globin transcription. In summary, our study reveals a model that nascent RNA sequesters PRC2 from its genome locus and blocks PRC2 activity in cis, which remodels local chromatin interaction and regulates neighboring gene expression. This provides an example of a robust non-autonomous gene-regulatory role for nascent RNAs in suppressing PRC2.

## Results

**Nascent transcripts interact with PRC2 and antagonize its chromatin binding.** We performed RNA Immunoprecipitation Sequencing (RIP-seq) in K562 cells for endogenous EZH2, which is the enzymatic subunit and the subunit with the highest RNA-binding affinity in PRC2[8]. The two well-established PRC2-interacting lncRNAs, XIST[2] and KCNQ1OT1[19], were both consistently enriched in our biological replicates (Supplementary Fig. 1a, b), validating our EZH2 RIP-seq. 1528 high-confidence EZH2-interacting RNA fragments were reproducibly identified using a stringent peak-calling algorithm (Supplementary Fig. 2a). Annotation of these fragments indicates no preference for transcripts generated from coding genes, non-coding genes, or intergenic regions (Fig. 1a). Detailed feature annotation of the fragments from the annotated gene loci further reveals that EZH2 preferentially binds intronic rather than exonic RNAs, disregarding the coding potential of the corresponding gene (Fig. 1a). To further study the identity of these EZH2-interacting RNAs, we examined their cellular localization by quantifying and comparing their normalized number of reads from nuclear and cytoplasmic-fractionated RNA-seq in K562 cells. 89% of these EZH2-bound RNA fragments have higher read counts in nuclear than in the cytoplasmic fraction, suggesting their nuclear localization (Fig. 1b). Further comparison of the RNA-seq read counts between nuclear polyA+ and nuclear polyA− fractions suggests more than 75% of these EZH2-interacting fragments are non-poly-adenylated (Fig. 1b). Collectively, these results support previous findings[6,12] that EZH2 predominantly binds nascent transcripts in the nuclei. Moreover, previous studies have suggested the selectivity of PRC2 toward guanine-rich RNAs[15,16]. To validate this in K562, we performed motif discovery analysis for EZH2-enriched RNA fragments with MeMe Suite[20]. One motif rich in guanine (G) and adenosine (A) was significantly enriched from our motif analysis (Supplementary Fig. 2b), highly agreeing with previous findings[15]. Collectively, our EZH2 RIP-seq in K562 cells confirmed that EZH2 prefers to bind G/A-rich nascent transcripts, similar to in other systems.

To further study how RNA affects PRC2 chromatin occupancy biochemically in K562, we inhibited global transcription by treating the cells with the RNA Polymerase II (Pol II) inhibitor 5,6-dichloro-1-β-D-ribofuranosylbenzimidazole (DRB) in culture and performed co-immunoprecipitation against PRC2 subunits EZH2 or SUZ12 (Supplementary Fig. 2c). While lack of RNA did not affect the binding of EZH2 or SUZ12 to other PRC2 subunits (SUZ12/EZH2 and RBBP4), both EZH2 and SUZ12 showed increased interaction with lysine 27 tri-methylated histone H3 (H3K27me3) and, to a lesser extent, with total H3 upon transcription blockage (Supplementary Fig. 2c). In line with this observation, the depletion of RNA by RNase A treatment also markedly enhanced PRC2 interaction with H3 and H3K27me3 (Supplementary Fig. 2d). Of note, both DRB and RNase A treated K562 cells showed a global increase in H3K27me3 relative to total H3 in the input samples, revealing that the enhanced PRC2 binding to the chromatin further led to increased histone methylation levels, supporting previous reports in mESC[9,12]. Collectively, these results support previous findings that RNA antagonizes PRC2 chromatin binding and thereby inhibits its methyltransferase activity in K562 cells.

**PRC2 binds a non-coding RNA LEVER upstream of the β-globin cluster.** Our EZH2 RIP-seq enriched many uncharacterized RNAs transcribed from unannotated intergenic regions (Fig. 1a). Among these uncharacterized RNAs, we identified one long transcript (which we termed LEVER) whose transcription start site locates 236 kb upstream of the β-globin gene cluster on

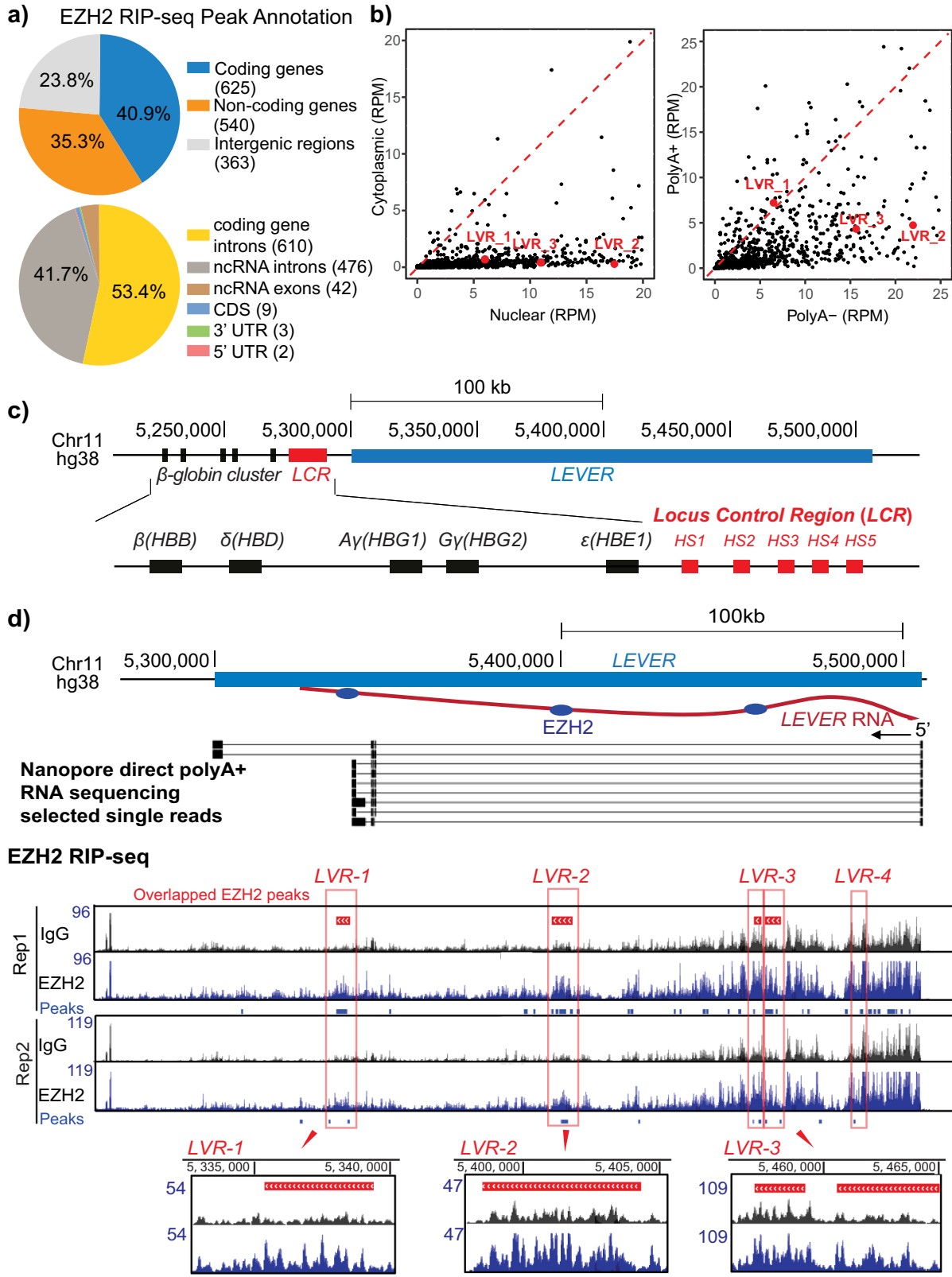

chromosome 11 (Fig. 1c). Multiple transcript fragments from this locus were highly enriched by RIP-seq (Supplementary Fig. 2a), and these fragments were further combined into four EZH2 overlapped peaks (Fig. 1d). The enrichment of *LEVER* RNA fragments was further validated by RIP-qPCR in three independent biological replicates (Supplementary Fig. 2f). Notably, *LEVER* RNA interacts with EZH2 to a comparable extent as that

of *XIST*, a well-established EZH2-interacting lncRNA, suggesting *LEVER* as a genuine EZH2-interacting RNA.

To examine if *LEVER* transcripts undergo splicing and to identify *LEVER* exons, if any, we performed long-read Nanopore direct sequencing of K562 total polyA+ RNA (Fig. 1d) as well as Nanopore cDNA sequencing of K562 nuclear RNA (Supplementary Fig. 2e). The Nanopore sequencing detected the entire

**Fig. 1 EZH2 binds a nascent transcript at *LEVER* locus. a** EZH2 preferentially binds intronic rather than exonic RNAs. Upper panel: Percentage of EZH2 RNA Immunoprecipitation Sequencing (RIP-seq) enriched RNA fragments annotated to coding genes, non-coding genes, or intergenic regions. The number of identified peaks annotated to each category are labeled in the brackets in the pie-chart legends. Lower panel: Percentage of EZH2 RIP-seq enriched RNA fragments originated from coding or non-coding genes annotated to different gene features. UTR, untranslated regions; CDS, coding sequences; ncRNA, non-coding RNA. **b** EZH2 predominantly binds nuclear polyA− nascent transcripts. Left panel: Scatter plot showing normalized read counts in nuclear (x-axis) and cytoplasmic (y-axis) fractioned RNA-seq for each EZH2 RIP-seq enriched RNA fragment. The read counts are normalized to total mapped reads as RPM (reads per million mapped reads). Enrichment of dots below the labeled diagonal reveals higher RNA-seq reads in the nuclear than in the cytoplasmic RNA fraction for most of the EZH2 RIP-seq fragments. Fragments annotated to *LEVER* as described in **d** are labeled in red. Right panel: similar as in left, except that the read counts of the identified RNA fragments are quantified in the nuclear polyA− (x-axis) and nuclear polyA+ (y-axis) fractions. **c** Chromosome map of the *LEVER* locus and the *β-globin* cluster. Transcription of *LEVER* (in blue) and the globin genes (in black) proceeds in a right to left direction. HS1 through HS5 (in red) correspond to the DNase I hypersensitive regions that comprise the *β-globin* locus control region (LCR). **d** EZH2 RIP-seq and Nanopore direct RNA sequencing of total polyA+ RNA in K562 cells demonstrating EZH2 binding intronic RNAs from the *LEVER* locus. In the Nanopore track, each line shows one single read from the direct polyA+ RNA sequencing: the bars represent the genomic regions captured by actual Nanopore read signals, while the lines in between those bars represent the genomic regions in between the captured signals from the same read, thereby suggesting splicing junctions. In the EZH2 RIP-seq tracks, the EZH2-enriched RNA peaks of each replicate (blue bars below the signal tracks) are called by ≥1.7-fold read coverage of EZH2 over IgG with Poisson distribution p-value ≤10$^{-3}$. Peaks generated from two biological replicates are further overlapped and merged as Overlapped EZH2 peaks, shown by the red bars above the RIP-seq tracks. Selected overlapped peaks (LVR-1, LVR-2, and LVR-3) are highlighted in red boxes with their read-coverage tracks shown in a zoomed-in view. The blue ovals in the top diagram represent the relative positions of EZH2 binding at the LVR-1, LVR-2, and LVR-3 regions.

spliced *LEVER* transcripts in single long reads, reflecting that the over 200 kb locus is transcribed from start to end into one nascent transcript. Importantly, the EZH2-RIP-enriched *LEVER* RNA fragments (i.e., LVR1-4) do not overlap with the spliced exons identified by our Nanopore polyA+ RNA sequencing. Therefore, we confirmed that EZH2 binds multiple intronic rather than exonic RNAs of the long *LEVER* nascent transcript (Fig. 1d). The nascent origin of these PRC2-interacting *LEVER* fragments was further supported by their enrichment in the nuclear and polyA− fraction as suggested by the fractionated RNA-seq analyses (Fig. 1b) as well as fractionated digital droplet RT-PCR (Supplementary Fig. 2g).

**LEVER RNA negatively regulates ε-globin.** Considering the genomic proximity of *LEVER* to the *β-globin* cluster (Fig. 1c), we studied the relationship between the expression levels of *LEVER* and *β-globin* cluster genes. To investigate the potential biological function of *LEVER* RNA in globin regulation, we first performed *LEVER* knock-out using the CRISPR-Cas9 system in K562 cells. We designed single guide RNAs targeting the *LEVER* promoter region to generate small deletions that disrupt transcription initiation at the *LEVER* locus. We obtained a *LEVER*-silenced clone with a deletion (270 bp) spanning the *LEVER* TSS site (Supplementary Fig. 3a) and confirmed that *LEVER* expression is barely detectable in this clone (Fig. 2a). Of note, in parental unmodified K562 cells, the fetal form of globin (Gγ-globin and Aγ-globin, together as γ-globin) and the embryonic form globin (ε-globin) genes were dominantly expressed, while the adult globins (β-and δ-globins) are undetectable by RT-qPCR. Interestingly, the mRNA level of ε-globin gene, which locates closest to the *LEVER* locus among the *β-globin* cluster genes (Fig. 1c), was upregulated upon the depletion of *LEVER* RNA (Fig. 2a). In contrast, the mRNA levels of fetal or adult globin genes were not significantly influenced: we did not detect a significant change in γ-globin by RT-qPCR (Supplementary Fig. 3c, left panel), while β- and δ-globins remained undetectable after *LEVER* knock-out.

To validate the observations made in the *LEVER* knock-out cells as well as to avoid potential clonal effects in the CRISPR knock-out system, we further constructed a doxycycline-inducible *LEVER* knock-down K562 line using the CRISPR interference (CRISPRi) system with a dead Cas9 (dCas9, whose endonuclease activity has been removed). Notably, the dCas9 we used in our CRISPRi[21] experiments is not fused with any epigenetic silencer, e.g., KRAB[22]. Similar to the *LEVER* knock-out cells, ε-globin

expression increased after *LEVER* knock-down (Supplementary Fig. 3b) without disturbing the expression of other *β-globin* forms (Supplementary Fig. 3c, right panel). Consistent with the increased RNA levels, increased RNA Pol II occupancy was detected at ε-globin promoter upon inducible *LEVER* knock-down (Fig. 2b). The increase of ε-globin expression was further confirmed at the protein level by western blot in both *LEVER* knock-out and knock-down cells (Fig. 2c, Supplementary Fig. 4a). Moreover, we further withdrew the doxycycline in the inducible CRISPRi knock-down system to restore *LEVER* RNA expression. As a result, the ε-globin activating effect was mitigated correspondingly to the restoration of *LEVER* RNA (Fig. 2d, Supplementary Fig. 3d). These ε-globin regulating effects observed upon *LEVER* interferences suggest *LEVER* RNA as a repressor of ε-globin transcription.

Furthermore, as ε-globin is expressed almost exclusively in the early human embryo under hypoxic conditions, we hypothesized that hypoxia may interfere with ε-globin expression and that *LEVER* could be potentially involved in this process. To test this, we cultured K562 cells in a 1% oxygen hypoxic environment. As a result, we observed increased ε-globin expression over time, and importantly, *LEVER* expression showed a corresponding decrease under these conditions (Fig. 2e, f, Supplementary Fig. 4b). To further validate the regulatory relationship between *LEVER* and ε-globin identified in the K562 system, we also analyzed the expression levels of *LEVER* and ε-globin in a single-cell RNA-seq data set of human-induced pluripotent stem cell (iPSC)-derived erythroblasts (Supplementary Fig. 3e), as well as in a panel of leukemia and lymphoma cell lines that express ε-globin (Supplementary Fig. 3f)[23]. A similar anti-correlation between *LEVER* and ε-globin was manifested as in the K562 cells, and this correlation was further supported by RT-qPCR of HEL and TF-1 cell lines (Supplementary Fig. 3g).

**PRC2 mediates the repressive effect of LEVER RNA on ε-globin.** As *LEVER* RNA was originally identified as a robust binding partner of PRC2 and thus can potentially influence PRC2 function, we were interested in determining if ε-globin repression by *LEVER* RNA is mediated by PRC2. To study the role of PRC2 in ε-globin transcriptional regulation, we knocked out the PRC2 enzymatic subunit EZH2 in K562 cells. The global H3K27me3 level, which is the enzymatic output of EZH2, decreased to a non-detectable level by western blot after EZH2 knockout (Fig. 3b, Supplementary Fig. 4c). Surprisingly, ε-globin expression was reduced in EZH2 knock-out

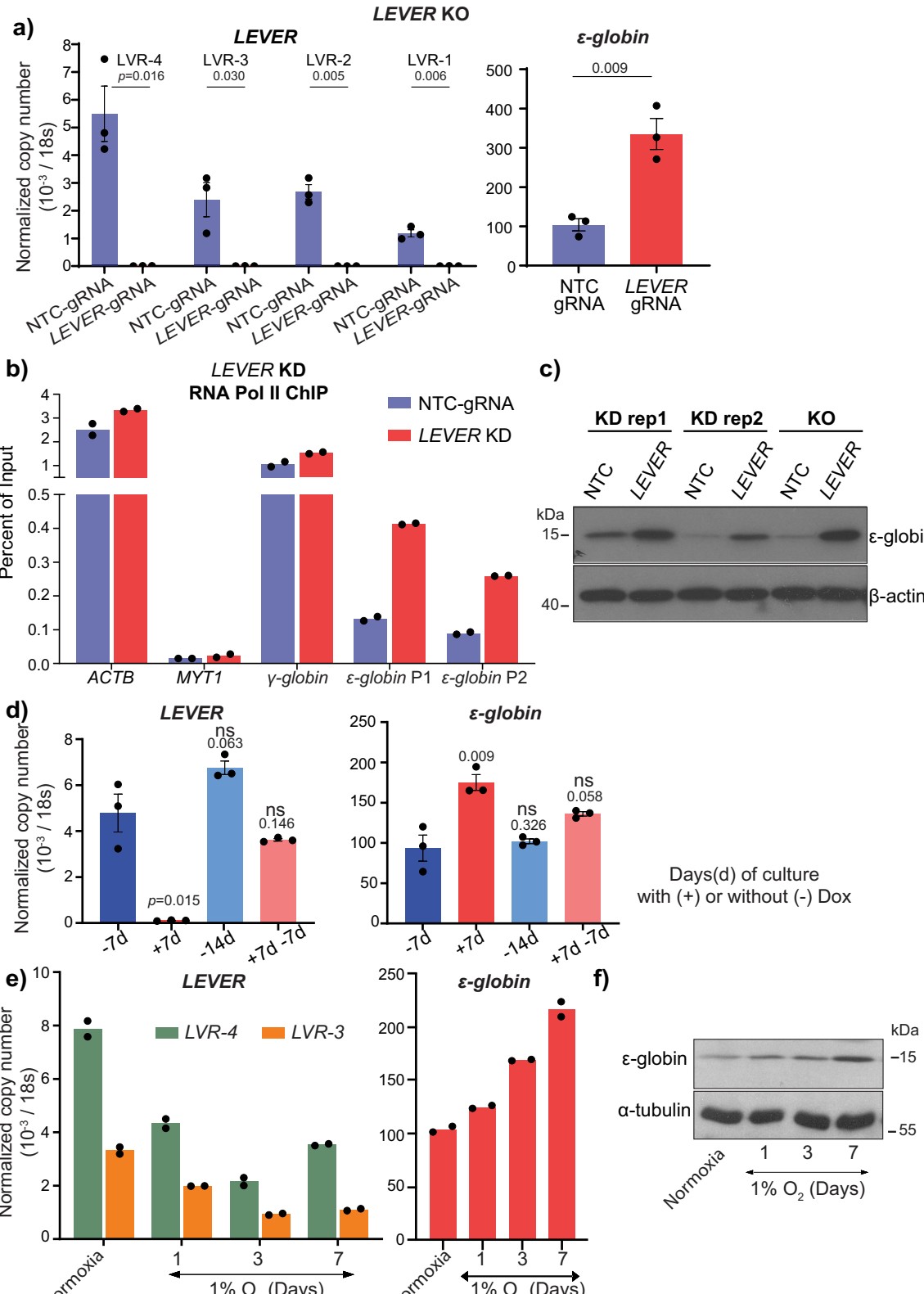

cells (Fig. 3a, b). Re-expression of wild-type EZH2 and its activating mutant (EZH2$^{Y641F}$)[24,25] in the knock-out cells restored $\varepsilon$-globin expression, while the truncated EZH2 mutant without methyltransferase motif (EZH2$^{\Delta SET}$) failed to do so (Fig. 3a, b). These results suggest that the PRC2 methyltransferase function, which generally represses the genome, maintains $\varepsilon$-globin expression level in the K562 cells.

Knowing that RNA-binding can inhibit PRC2 function (Supplementary Fig. 2c, d) and that PRC2 maintains $\varepsilon$-globin expression (Fig. 3a, b), we hypothesized that LEVER binding to PRC2 may inhibit PRC2 function and thereby down-regulate $\varepsilon$-globin expression. To test this hypothesis, we overexpressed two LEVER RNA fragments encompassing the consensus EZH2-bound guanine-rich motif (Supplementary Fig. 2b) within the LVR-1 and LVR-2 regions

**Fig. 2 *LEVER* RNA negatively regulates ε-globin. a** *LEVER* and ε-*globin* expression after *LEVER* knock-out (KO) in K562 cells measured by RT-qPCR. *LEVER* RNA expression is measured by primers detecting four selected regions (marked in Fig. 1d) in non-targeting control (NTC-gRNA) or *LEVER* knock-out (*LEVER*-gRNA) K562 cells. Data from n = 3 biological replicates are shown. Error bars represent SEM. Statistical difference was calculated using Welch's one-tailed *t*-test. **b** Chromatin Immunoprecipitation qPCR (ChIP-qPCR) of RNA Pol II at ε-*globin* and γ-*globin* promoters in non-targeting control (NTC-gRNA) and *LEVER* knock-down (*LEVER* KD) cells. *ACTB* and *MYT1* serve as Pol II ChIP positive and negative controls, respectively. Pol II occupation at ε-*globin* promoter is measured by two primer sets (ε-*globin* P1 and ε-*globin* P2). Data from technical replicates of one biological sample is shown. **c** ε-*globin* protein abundance in *LEVER* KO or KD K562 cells measured by western blot. β-actin serves as the loading control. **d** *LEVER* and ε-*globin* expression in inducible *LEVER* KD K562 cells with (+) or without (−) doxycycline treatment for the indicated duration. *LEVER* expression is quantified at the LVR-4 region. Data from n = 3 biological replicates are shown. Error bars represent SEM. Statistical difference was calculated using Welch's one-tailed *t*-test comparing with the −7d sample. ns, not significant. **e** Time course RT-qPCR analysis of *LEVER* RNA and ε-*globin* in K562 cells during 7-day hypoxic culture under 1% oxygen. Data from technical replicates of one biological sample is shown. **f** ε-globin protein abundance in K562 cells cultured under 1% oxygen. α-tubulin serves as the loading control.

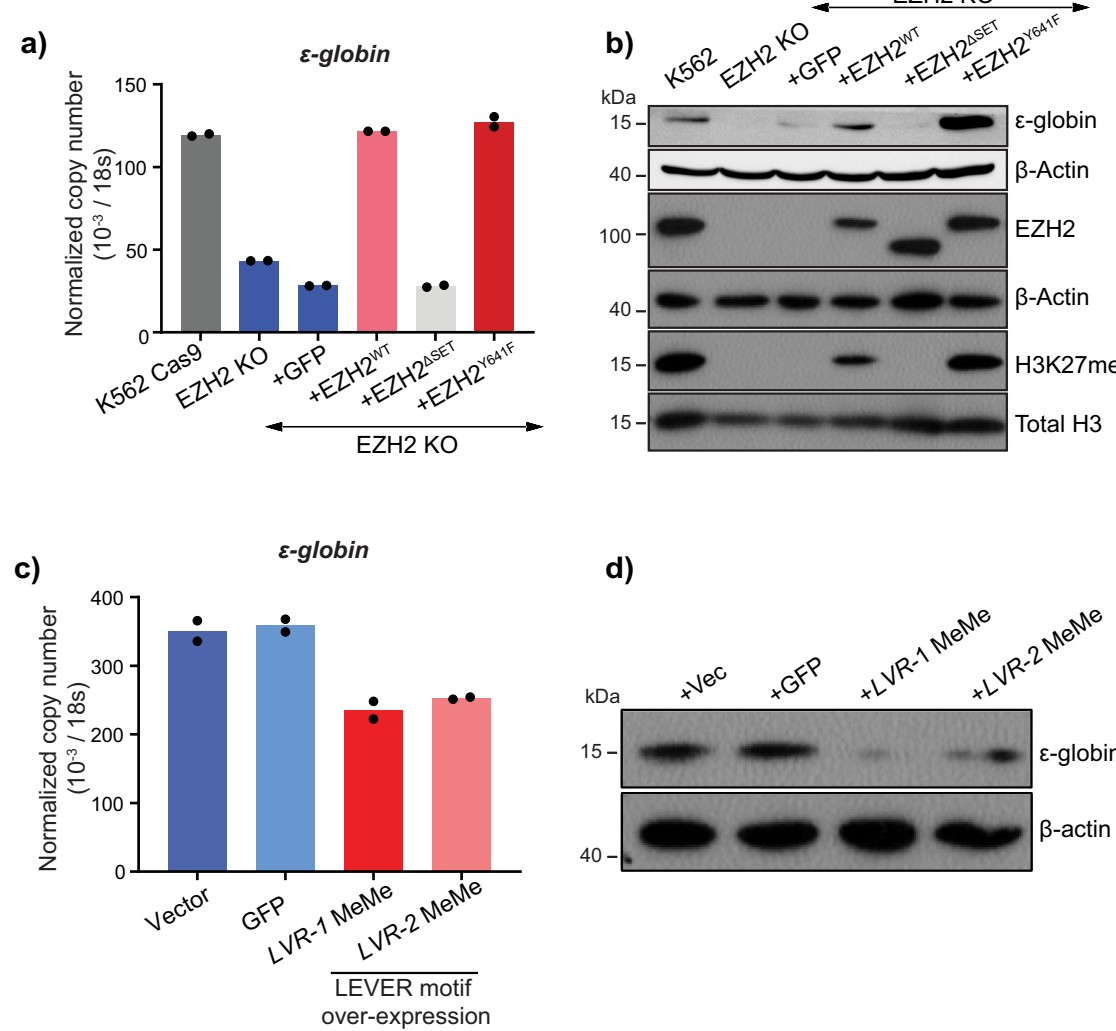

**Fig. 3 PRC2 enzymatic function maintains ε-*globin* expression in K562. a** RT-qPCR analysis of ε-*globin* in K562 Cas9-expressing parental cells, EZH2 knock-out (EZH2-KO) cells, and EZH2-KO cells rescued with GFP, EZH2, EZH2$^{\Delta SET}$, and EZH2$^{Y641F}$. EZH2$^{\Delta SET}$, truncated EZH2 without methyltransferase enzymatic domain; EZH2$^{Y641F}$, EZH2 gain of methyltransferase enzymatic function mutation. The EZH2-KO cells were transduced with lentivirus to stably express these constructs. Data from technical replicates of one biological sample is shown. **b** Western blot analysis of ε-*globin*, EZH2, and global H3K27me3 in K562 wild type, EZH2 KO, and rescue cells as in **a**. β-actin and total histone H3 were used as loading controls. **c, d** ε-*globin* expression in *LEVER* KO K562 cells rescued by LVR-1 MeMe or LVR-2 MeMe. LVR-1 and LVR-2 MeMe represent RNA fragments encompassing MeMe enriched EZH2-interacting RNA motif (Supplementary Fig. 2b) within the LVR-1 and LVR-2 regions (Supplementary Fig. 5a). *LEVER* KO K562 cells were transduced with empty vector, GFP RNA, LVR-1 MeMe, or LVR-2 MeMe expressing lentivirus and analyzed for ε-*globin* expression using **c** RT-qPCR and **d** western blot. For (**c**), Data from technical replicates of one biological sample is shown.

(termed LVR-1 MeMe and LVR-2 MeMe, respectively, Supplementary Fig. 5a) independently in the *LEVER* KO cells (Supplementary Fig. 5b). In short, we selected ~300 nt G/A rich fragments from the RIP enriched peaks of LVR-1 and LVR-2. As the G/A preference of PRC2 has been validated by previous publications from different groups[15,16], we hypothesized that these selected fragments with the validated motif should have stronger PRC2 binding affinity than other regions within the whole *LEVER* transcript. Meanwhile, we used a non-translatable non-G/A-rich EGFP RNA fragment (~700 nt) as a negative control. Indeed, we found that re-expression of the EZH2-interacting *LEVER* fragments can reverse the *ε-globin* activating effect caused by the loss of the whole *LEVER* transcript. Changes in *ε-globin* expression were demonstrated at both *ε-globin* mRNA (Fig. 3c) and protein levels (Fig. 3d, Supplementary Fig. 4d). In summary, the above data suggest that PRC2 enzymatic function maintains *ε-globin* expression, while *LEVER* RNA can down-regulate *ε-globin* through inhibiting PRC2.

**LEVER RNA sequesters PRC2 and inhibits H3K27 methylation at the LEVER locus.** To validate whether endogenous *LEVER* RNA inhibited PRC2-mediated epigenetics, we performed chromatin immunoprecipitation assays for EZH2 and histone marks in the *LEVER* knock-out and knock-down cells. Indeed, in the *LEVER* knock-out cells, we observed increased EZH2 occupancy at the *LEVER* locus (Fig. 4a), which was accompanied by markedly elevated H3K27me3 levels at the same loci (Fig. 4b), suggesting that *LEVER* RNA sequesters PRC2 and inhibits H3K27 methylation at the *LEVER* locus. In contrast, no noticeable increase in either EZH2 or H3K27me3 was detected at *MYT1* (serving as a positive control for EZH2 and H3K27me3 ChIP) or *CCDC26* (negative control) regions (Fig. 4a, b). Importantly, this suggests that the PRC2-related chromatin changes at the *LEVER* locus are not likely to be global effects. Consistently, inducible knock-down of *LEVER* by Pol II blockade similarly changed PRC2-related epigenetic features along the *LEVER* locus: histone ChIP-seq showed increased H3K27me2/me3 levels restrictedly at the *LEVER* locus (Fig. 4c, Supplementary Fig. 6a) but not at the adjacent loci, suggesting that loss of *LEVER* enhanced PRC2 activity and reduced chromatin accessibility. In addition, H3K27 acetylation (H3K27ac) at the *LEVER* locus was correspondingly reduced, further supporting the above observations. These ChIP-seq results were further validated at selected regions by qPCR (Supplementary Fig. 6b–d). The results conform to the global PRC2 inhibiting effects caused by the global loss of nascent transcripts (Supplementary Fig. 2c, d), indicating that loss of *LEVER* nascent transcript enhanced local PRC2 occupancy and its progressive H3K27 methylation products on the local chromatin. Besides, we observed a positive correlation between *ε-globin* expression (Fig. 3a, b) and H3K27me3 levels within the *LEVER* locus (Fig. 4d) in EZH2 knocked-out and rescued cells, which suggests the importance of PRC2-mediated epigenetics within the *LEVER* locus in regulating *ε-globin* expression.

**The LEVER locus regulates ε-globin through long-range chromatin interactions.** While we have demonstrated that *LEVER* RNA inhibits EZH2 function at its own locus in cis, we wonder how these local epigenetic changes could lead to the repression of downstream *ε-globin* gene. Interestingly, we found that the genomic locus of *LEVER* overlaps with a large genomic domain that was reported to interact with *β-globin* locus control region (LCR)[26]. Therefore, we asked whether the epigenetic changes within the *LEVER* locus could remodel local chromatin interactions and thereby influence *ε-globin* expression. To test this hypothesis, we performed 4C-seq analysis before and after *LEVER* promoter knock-out to compare local chromatin interaction changes upon *LEVER*

RNA loss. We positioned two viewpoints at the *ε-globin* promoter and another two viewpoints at HS3 and HS4 regions within the *β-globin* LCR, which are well-established activating elements of the *β-globin* cluster[27] (Fig. 5). Consistent with the previous finding[26], our 4C-seq analysis suggested that *LEVER* chromatin interacts with LCR in K562 cells with a few regions identified as reproducibly significant LCR interacting peaks, although the interaction was weaker compared with those between LCR and globins (Supplementary Fig. 7). Importantly, loss of *LEVER* RNA resulted in the reduction of such interaction (Fig. 5, Supplementary Fig. 7), presumably due to the compromised accessibility of *LEVER* locus caused by PRC2-mediated histone methylations. More importantly, the interaction between the *ε-globin* promoter and LCR regions was substantially enhanced upon *LEVER* KO, as supported by the results from both viewpoints (Fig. 5). Together with the increased Pol II occupancy at the *ε-globin* promoter (Fig. 2b), our results indicate that the enhanced interaction between *ε-globin* promoter and LCR, which is resulted from *LEVER* RNA loss, further contributes to the *ε-globin* transcriptional activation. Since the *LEVER* locus interacts with LCR, we propose that it may work as a competitor of the *ε-globin* promoter for LCR binding, thereby exerting its negative regulatory effect on *ε-globin* expression. Collectively, our 4C-seq results from different viewpoints support that PRC2-mediated local epigenetic changes at *LEVER* locus can influence *ε-globin*-LCR interaction and thus affects *ε-globin* expression.

**Discussion**
Following the initial discovery of the specific PRC2-lncRNA interaction and its supportive effect on PRC2 functions at PRC2 target genes[2–4,19], more pieces of evidence revealed a promiscuous PRC2-nascent RNA interaction that inhibits local PRC2 function[5,7,8,12,14]. While previous studies of the latter viewpoint were performed either in vitro or exclusively in the embryonic stem cell systems, our study provides further evidence to support this model. In summary, we demonstrated that PRC2 manifests genome-wide interaction with nuclear-localized, non-polyA tailed, guanine-rich intronic RNAs that are transcribed from active gene loci, and we further confirmed that global transcription inhibition as well as RNA degradation increase PRC2 interactions with histone H3 and global H3K27me3 levels. Following the global analysis, we further identified and characterized the *LEVER* locus, which generates PRC2-binding nascent transcripts near the *β-globin* cluster. Using both CRISPR knock-out and inducible CRISPRi knock-down systems, we demonstrated that *LEVER* nascent RNA prevents PRC2 from occupying the chromatin and thereby curtails EZH2 methyltransferase activity at the *LEVER* locus in cis. These observations made at a single gene locus support the model demonstrating the inhibition of PRC2 by nascent transcripts.

Moreover, while such PRC2-inhibitory interaction with nascent RNAs is becoming clear, its biological meaning remains to be explored. In our work, we reported that the autonomous PRC2 inhibition mediated by nascent RNAs can lead to changes in chromatin interactions and expression of neighboring genes. To summarize our proposed model (Fig. 6), in the presence of the *LEVER* nascent RNA, the *LEVER* locus competes with the *ε-globin* promoter to bind with the LCR, a well-established enhancer region for the *β-globin* cluster[27,28], and thereby suppresses *ε-globin* transcription. In the absence of *LEVER* nascent RNA, the released PRC2 occupies the *LEVER* chromatin, facilitates H3K27 methylation, and reduces the accessibility of the *LEVER* locus. This, in turn, permits the interaction between the LCR and the *ε-globin* promoter, promoting transcription of *ε-globin*. In sum, we demonstrated that the *LEVER* nascent RNA can activate the *LEVER* locus that potentially works as a negative

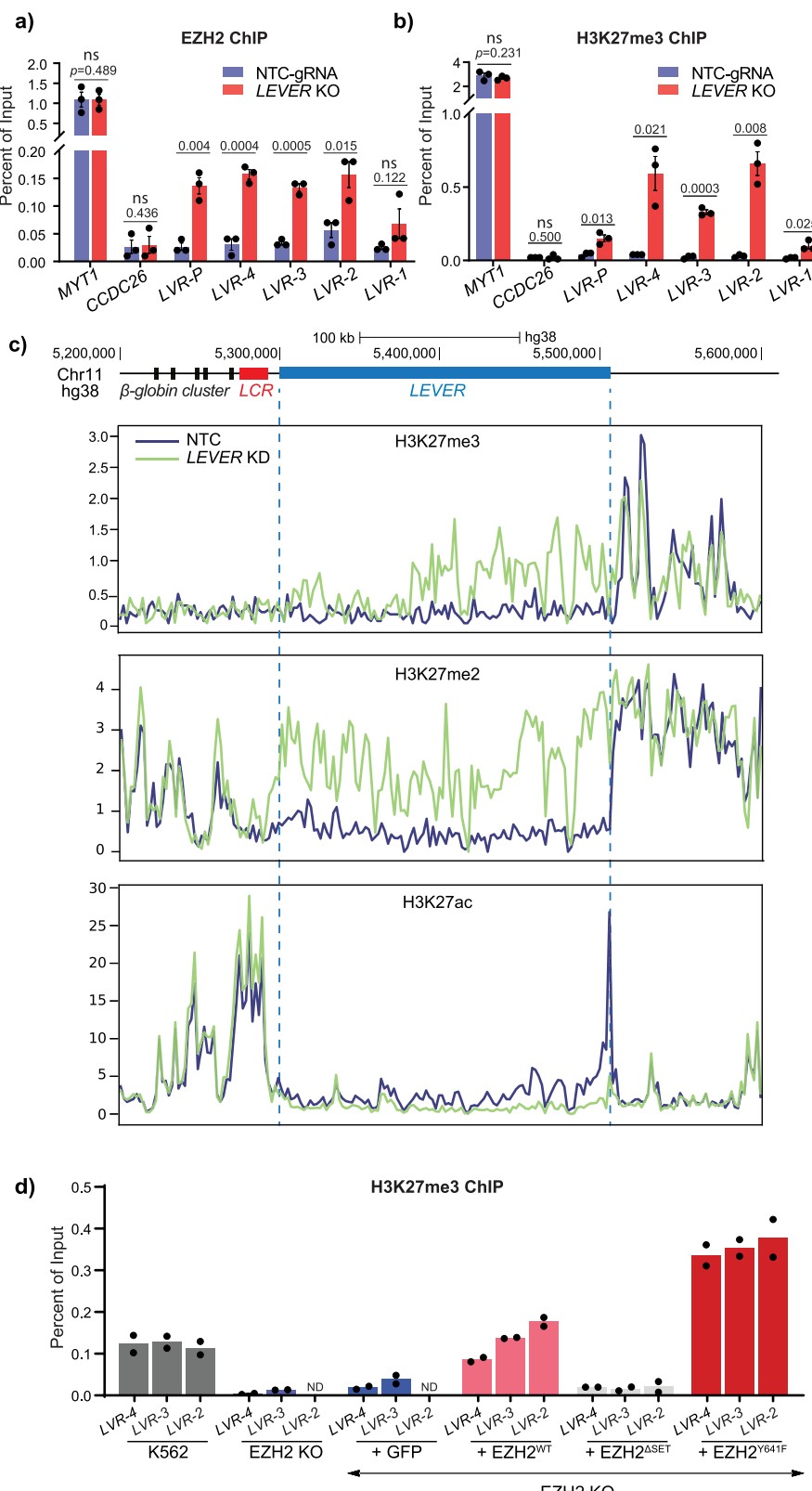

**Fig. 4 *LEVER* RNA sequesters EZH2 and blocks histone H3K27 methylation at the *LEVER* locus in cis. a, b** EZH2 (**a**) or H3K27me3 (**b**) chromatin occupation at selected regions within the *LEVER* locus. *MYT1* and *CCDC26* serve as ChIP positive and negative controls, respectively. LVR-P primer pair is designed at the *LEVER* promoter region. Data from *n* = 3 biological replicates are shown. Error bars represent SEM. Statistical difference was calculated using Welch's one-tailed *t*-test. ns, not significant. **c** H3K27me3, H3K27me2, and H3K27ac ChIP-seq profiling at *LEVER* and its neighboring locus in dox-inducible non-targeting control (NTC) or *LEVER* knock-down (*LEVER* KD) K562 cells treated with dox for 28 days. **d** H3K27me3 ChIP-qPCR of EZH2-KO and rescued cells as in Fig. 3a, b at LVR-4, LVR-3, and LVR-2 regions.

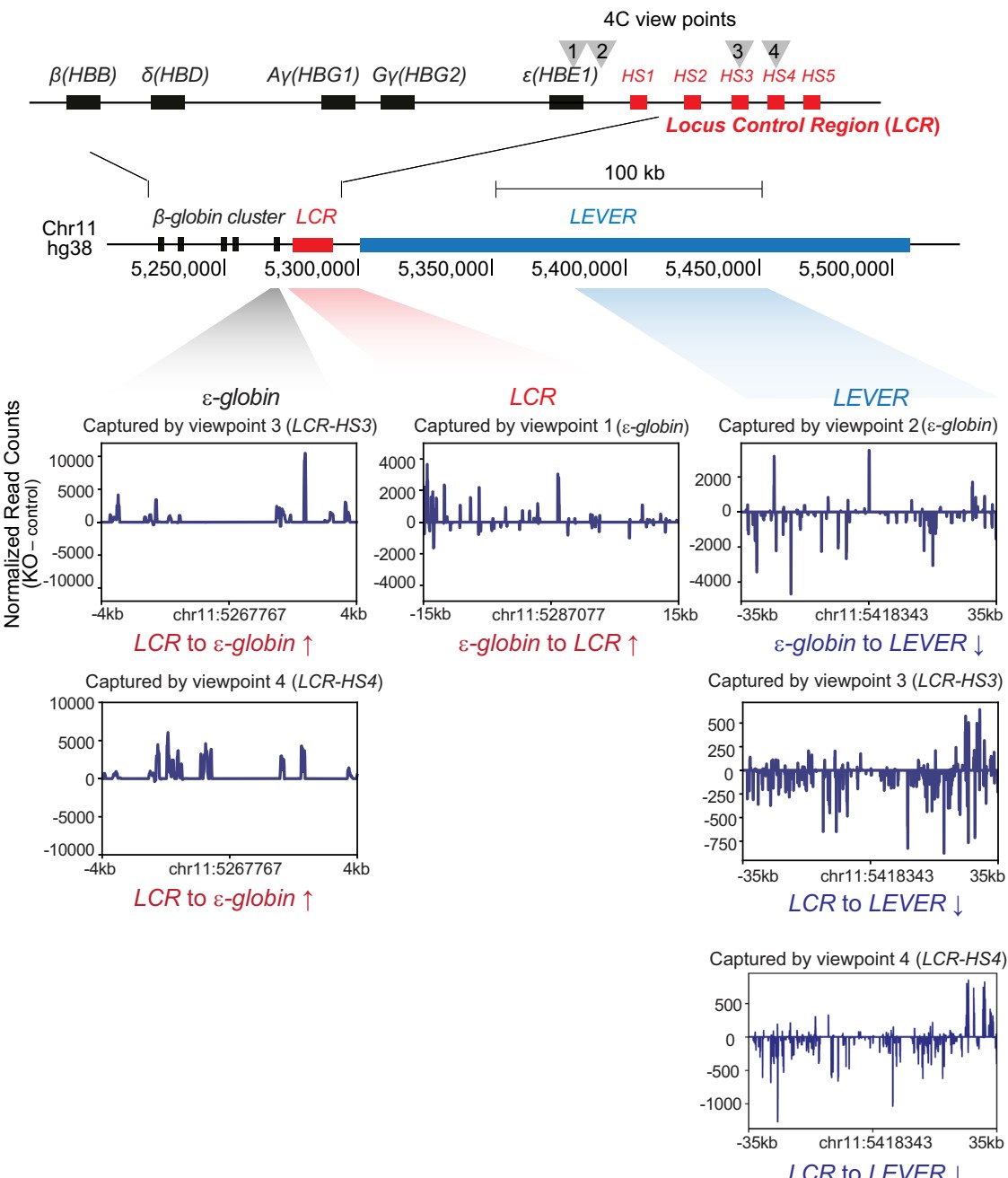

**Fig. 5 The *LEVER* locus regulates *ε-globin* through long-range chromatin interactions.** 4C analysis comparing local chromatin interactions near the *β-globin* cluster in *LEVER* KO and NTC cells shows increased *ε-globin*-LCR interaction upon *LEVER* knock-out. We positioned two viewpoints at the *ε-globin* promoter (viewpoint 1 and 2) and two viewpoints at HS3 (viewpoint 3) and HS4 (viewpoint 4) regions within the *β-globin* locus control region (LCR). Captured genomic fragments displayed in the windows correspond to the genomic regions shaded with gray (*ε-globin*), red (LCR), and blue (*LEVER*) colors below the chromosome map, with the central position of each region annotated below each window. Differentially interacted regions are shown in subtracted reads by comparing *LEVER* KO and NTC cells, thereby positive or negative values represent increased or decreased interactions upon *LEVER* KO, respectively.

regulatory element of *ε-globin* gene expression through chromatin looping.

Interestingly, we found that transcripts generated from similar genomic regions upstream of the *β-globin* cluster in the mouse genome were previously identified and annotated as polycomb-associated ncRNAs, similar to *LEVER*[29], although the two loci from the two species share poor sequence homology. This observation raises a question: whether the regulatory effects of the corresponding *LEVER* locus and its RNAs on the *β-globin* cluster could be functionally conserved in humans and mice.

Considering the promiscuous RNA-binding pattern of PRC2, some PRC2-binding non-coding RNAs may have evolved to become opportunistic regulatory elements that are epigenetically functional to neighboring genes and thereby conserved their relative position to neighboring protein-coding genes without the need to conserve their sequences in different species. The syntenic transcription of lncRNA has been proposed together with conserved sequence, structure, and function as the four dimensions of lncRNA conservation[30]. Our model in PRC2-nascent RNA interaction and its regulatory function on nearby

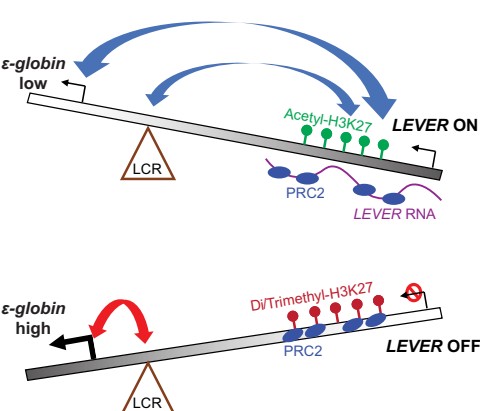

**Fig. 6 The proposed model of *LEVER* RNA regulating *ε-globin* transcription.** Top diagram: *LEVER* RNA sequesters and antagonizes PRC2 in cis. This maintains the accessibility of the *LEVER* locus and facilitates its chromatin interaction with the *ε-globin* promoter and LCR region, which negatively regulates *ε-globin* transcription. Bottom diagram: When *LEVER* is silenced, the LCR can now interact with the *ε-globin* promoter to induce *ε-globin* transcription.

genes may explain such synteny conservation observed in some lncRNAs.

Considering the reported effects of epigenetic processes in regulating transcription[31], the epigenetic and *ε-globin* expression changes observed in our CRISPR-based *LEVER* knock-out/knock-down systems could be caused in part by silencing *LEVER* transcription event. Due to the difficulty to knock down a nuclear-localized, very long (>200 kb) nascent transcript without disrupting its transcription through currently available RNA targeting systems, we tried but failed to reduce *LEVER* nascent RNA abundance using shRNA or Cas13a[32]. Therefore, the independent effects of *LEVER* RNA and transcription remain to be resolved in future studies. This question could be addressed by a localized overexpression experiment with techniques enabling localized enrichment of *LEVER* fragments at the *LEVER* locus, similarly as designed by Beltran et al.[16]. Such a method would further demonstrate the causal relationships between PRC2-regulated epigenetics, chromatin interaction, and *ε-globin* expression proposed in our model (Fig. 6). While our investigation and proposed model focus on the *cis*-regulating role of the *LEVER* nascent transcript based on the assumption that the nascent transcript is tethered to its own locus[33,34], we can not rule out the possibility that *LEVER* RNA, either the nascent transcript or the spliced mature forms, can also act in trans. Finally, we used K562 cell line as our model system to study the molecular mechanism of PRC2-*LEVER* interaction in regulating *ε-globin* expression. The cell line was chosen due to its feasibility for genetic modifications and its culturability to obtain sufficient materials for ChIP and 4C assays that require large numbers of cells for reliable results. However, the exact physiological importance of *LEVER* in regulating *ε-globin* expression will require future studies in more physiologically relevant systems, e.g., in human hematopoietic stem and progenitor cells.

## Methods
**Cell culture**. K562 and HEK293T cell lines were obtained from the American Type Culture Collection (ATCC) and were used in experiments at their early passages. K562 cells were cultured in RPMI (Biowest) supplemented with 10% fetal bovine serum (FBS) (Biowest). HEK293T cells were cultured in DMEM supplemented with 4500 mg/L glucose (Biowest), 2 mM L-glutamine (Thermo Scientific), 1X MEM non-essential amino acids (Thermo Scientific), 1 mM sodium pyruvate, and 10% fetal bovine serum (FBS) (Biowest). All cell lines were tested for mycoplasma every 6 months.

**Lentivirus production**. pCMV-dR8.91, pCMV-VSV-G, and lentivectors were transfected into 10 million 293T cells using Lipofectamine 2000 following the manufacturer's instructions. Supernatants were harvested at 48 and 72 h after transfection. Viruses were concentrated 100–300 times through centrifugation after filtering through 0.45 μM syringe filters and stored at −80 °C. The titers of viruses were determined on HeLa cells using flow cytometry. The plasmids used in this study are listed in Supplementary Data 1.

**CRISPR/Cas9 and CRISPRi/dCas9**. K562 cells were transduced with the FUCas9Cherry or FUdCas9Cherry lentiviruses at 0.5 MOI with 5 μg/mL polybrene. The transduced cells were sorted for mCherry positive as K562-Cas9/dCas9 stable lines. FgH1tUTG-sgRNA viruses were further transduced into these K562-Cas9/dCas9 stable lines and sorted for GFP positives. To generate EZH2 KO, in vitro synthesized sgRNA (Synthego) was transiently transfected into K562-Cas9 cells with DharmaFECT 1 (DHARMACON) following the manufacturer's instructions. Of note, the DNA editing efficiency of the sgRNAs at their specific targeting regions was determined in K562-Cas9-sgRNA cells by the T7 endonuclease I (NEB) assay (as described in the manufacturer's protocol) after sgRNA induction by 1 μg/mL doxycycline for 3 days. EZH2-sgRNA transfected (48 h) K562-Cas9 cells or Induced K562-Cas9-sgRNA cells were seeded into single-cell clones in 96-well plates. Successful KO clones screened by RT-qPCR (for *LEVER* KO) or western blot (for EZH2 KO) were expanded and analyzed by genotyping. The list of sgRNAs used can be found in Supplementary Data 1.

**Ectopic EZH2 or *LEVER* expression**. For the ectopic expression of EZH2 and its mutants, N-terminal Flag-tagged EZH2 (WT) was cloned from K562 cDNA into a modified pLX304 plasmid in which the CMV promoter was replaced with the CAG promoter. The EZH2$^{\Delta SET}$ and EZH2$^{Y641F}$ mutants were derived from EZH2 WT and cloned into the same vector. For the ectopic expression of *LEVER* fragments, LVR-1 or LVR-2 were amplified from K562 genomic DNA and cloned into a modified pLX304CAG plasmid, in which the CAG promoter and the hPGK promoter were inverted, and a BGH stop sequence was added to ensure that only the designated *LEVER* RNA fragments are being expressed. Lentivirus was produced as stated above and transduced into corresponding knock-out K562 cells and selected with blasticidin.

**Western blot**. Protein lysates were prepared in RIPA buffer (50 mm Tris-Cl pH 7.5, 150 mM NaCl, 1 mM EDTA, 1% NP-40, 0.5% sodium deoxycholate (NaDOC), 0.1% SDS), and the lysates were mildly sonicated into clear supernatants before subjected to protein quantification by the Pierce BCA assay (Thermo Scientific). 10–20 μg of protein lysate was loaded into each well of SDS-PAGE mini gel. The electrophoresis was performed in Tris-glycine buffer supplemented with 0.1% SDS. Proteins were then transferred onto nitrocellulose membranes (Bio-Rad) in Tris-glycine-methanol transfer buffer. The membrane was blocked by 5% skim milk/TBST and incubated with primary antibodies followed by HRP-conjugated secondary antibodies diluted in 5% skim milk/TBST. Chemiluminescence was detected by exposing X-ray film to the blot pre-incubated with the Luminata Crescendo Western HRP substrate (Millipore). The list of antibodies used can be found in Supplementary Table 1.

**Quantitative PCR (qPCR)**. One microgram of RNA was reverse transcribed using Qscript cDNA Supermix (QuantaBio). qPCR on cDNA or ChIP DNA was performed with GoTaq qPCR Master Mix (Promega) under the manufacturer's instruction on a QS5 system (Thermo Scientific). The copy numbers were measured by predetermined standard curves. The relative expression or enrichment to control samples was calculated based on the ΔΔCt method. Quantitative digital droplet PCR was performed on a QX200 Droplet Digital PCR (ddPCR) System (Bio-Rad) with EvaGreen Digital PCR Supermix (Bio-Rad) under the manufacturer's instructions. Primers used in this study are listed in Supplementary Data 1.

**RNA fractionation and sequencing**. Nuclear RNA purification was performed as described previously[35]. Briefly, 12 million cells were harvested and washed first with 20 mL of cold PBS supplemented with 5 mM ribonucleoside Vanadyl complex (NEB) and 1 mM phenylmethylsulphonyl fluoride (PMSF), and then with 20 mL of cold PBS supplemented with 1 mM PMSF. The cell pellet was lysed in 1.7 mL of cold hypotonic buffer (10 mM HEPES pH 7.6, 25 mM KCl, 0.15 mM spermine, 0.5 mM spermidine, 1 mM EDTA, 2 mM sodium butyrate, 1.25 M sucrose, 10% glycerol, 5 mg/mL BSA, 0.5% NP-40) supplemented with 1 mM PMSF, EDTA-free protease inhibitor cocktail (Roche) and 20 U/mL SUPERase-In (Invitrogen). The cells were homogenized with a 15 mL Dounce homogenizer (by 10 strokes with Pestle A and 45 strokes with Pestle B). The homogenized cells were diluted in 7.8 mL of NP-40 free hypotonic buffer containing 2 M sucrose. 3.2 mL of cushion buffer (BSA and NP-40 free hypotonic buffer containing 2 M sucrose) was added to the bottom of a polypropylene tube (Beckman Coulter, 331372), and the cell lysate was overlaid on the cushion buffer without disrupting the bottom phase. Nuclei were harvested after ultra-centrifugation at $100,000 \times g$ for 1 h at 4 °C with a SW41Ti rotor and Beckman Coulter TH641 ultra-centrifuge. The supernatant was removed, and the nuclear RNA was extracted by Trizol according to the

manufacturer's protocol. Nuclear polyA+ RNA was extracted with a NEBNext Poly(A) mRNA Magnetic Isolation Module according to the manufacturer's protocol, and the nuclear polyA− fraction was extracted from the unbound supernatant by ethanol precipitation. Cytoplasmic RNA was isolated with a Qiagen RNeasy kit according to the manufacturer's protocol. The fractionated RNA was subjected to DNase I digestion and then re-purified using Trizol. The RNA was quantified with the Qubit RNA HS assay kit (Thermo Scientific), and rRNA was removed by the Ribo-Zero Gold rRNA Removal Kit (Illumina). The stranded RNA-seq libraries were constructed with the Script-seq V2 RNA-seq library preparation kit (Illumina). The 250–500 bp size-selected libraries were sequenced on the Illumina MiniSeq platform with MiniSeq High Output 2 × 150 paired-end kit.

**DRB/RNase A treatment and immunoprecipitation.** For DRB treatment, 12 million K562 cells were treated with 100 μM DRB for 24 h before subjected to nuclei extraction. For RNase A treatment, 10 million K562 cells were first harvested and washed with 1X PBS, and then treated with 1 mg/mL RNase A in 0.05% Tween-20 for 10 min at RT before subjected to nuclei extraction. The DRB/RNase A treated cells were lysed on ice for 5 min with cell lysis buffer (20 mM Tris pH 7.5, 10 mM KCl, 1.5 mM MgCl$_2$, 0.1% Triton X-100) supplemented with EDTA-free protease inhibitor cocktail (Roche), phosphatase inhibitor (Roche), 1 mM DTT, and 1 mM PMSF and then centrifuged at 800 RPM for 10 min at 4 °C. The nuclei pellets were washed once with the cell lysis buffer without Triton X-100 and harvested by centrifugation at 1300 × g for 4 min at 4 °C. Nuclei were resuspended in resuspension buffer (50 mM Tris pH 7.5, 150 mM NaCl, 2 mM EDTA) supplemented with EDTA-free protease inhibitor cocktail (Roche), phosphatase inhibitor (Roche), 80 U/mL RNaseOUT (ThermoScietific), and 1 mM PMSF. The nuclei were sonicated using EpiShear Probe Sonicator (Active Motif) for 15 s at 30% amplitude. 500 μl of the lysate was incubated with EZH2 antibody (1:250), SUZ12 antibody (2 μg), or rabbit IgG isotype (2 μg) in the cold room for 12 h. The lysate was incubated with 50 μl pre-blocked dynabeads protein A (Invitrogen) for 2 h at 4 °C. The beads were washed with washing buffer (150 mM NaCl, 10 mM Tris-HCl, pH 7.4, 1 mM EDTA, 1 mM EGTA, pH 8.0, 1% Triton X-100, 0.5% NP-40, 1 U/mL SUPERase-In) for three times before eluted with 30 μl of RIPA buffer and 15 μl of 6X Laemmli buffer at 95 °C for 10 min. The eluted complex was analyzed with western blot.

**RNA immunoprecipitation (RIP).** RIP was performed as described previously[35] with the following modifications. (A) SUPERase-In was used in substitution of ribonucleoside Vanadyl complex from the cell lysis step. (B) Fixed K562 cells were lysed with Dounce homogenizer by 10 strokes using Pestle A and 60 strokes using Pestle B. (C)The nuclei were sonicated with Bioruptor (Diagenode) for 8 cycles (30 s on, 30 s off, high power). (D) Nuclear lysates were pre-cleared with Dynabeads protein G twice at 4 °C. (E) 5 μg of EZH2 or IgG isotype control antibodies were used in each RIP experiment. RIP experiments were performed in three biological replicates, and two of these replicates were independently sequenced. For sequencing, the immunoprecipitated RNA was quantified with the Qubit RNA HS assay kit (Thermo Scientific) before rRNA Removal with Ribo-Zero Gold rRNA Removal Kit (Illumina). 15 ng of the rRNA-depleted RNA was used for sequencing library construction. The stranded RIP-seq libraries were constructed with a Script-seq V2 RNA-seq library preparation kit (Illumina). The libraries were subjected to size selection (250–500 bp) on a 4–20% TBE PAGE gel (Thermo Scientific). The recovered libraries were sequenced on the Illumina MiniSeq platform using the MiniSeq High Output kit with 2 × 150 paired-end.

**Chromatin immunoprecipitation (ChIP).** ChIP was performed as described previously[36] with minimal modifications. Briefly, K562 chromatin was sonicated with Bioruptor (Diagenode) for 15–25 cycles (30 s on, 30 s off, high power), and immunoprecipitation was performed with Dynabeads protein G. For ChIP-sequencing, the libraries were constructed using a Thru-PLEX DNA-seq 12S kit (TakaraBio). The libraries were subjected to size selection (250–500 bp) on a 4–20% TBE PAGE gel (Thermo Scientific). The recovered libraries were sequenced on the Illumina Nextseq platform using Nextseq 500 high-output kit with 2 × 76 paired-end. For H3K27me3, H3K27me2, or H3K27Ac ChIP-seq, one replicate was sequenced, and the results were validated by ChIP-qPCR with independent biological replicates.

**Oxford Nanopore sequencing.** Fractionated nuclear RNA was used to construct a cDNA library using a 1D sequencing kit (SQK-LSK108) according to the manufacturer's instructions. In brief, first-strand cDNA was generated from 20 ng of rRNA-depleted nuclear RNA using SuperScript IV reverse transcriptase with 0.35 pmol/μl random hexamer at 52 °C for 60 min. Second strand cDNA was synthesized with the NEBNext second strand synthesis kit for 1 h at 16 °C. cDNA was purified with AMPure XP bead. Library was constructed with LongAmp Taq PCR for 18 cycles. The PCR product was purified with phenol:chloroform:isoamyl alcohol (25:24:1) and ethanol precipitated in the presence of glycogen and was finally dissolved in 10 mM Tris buffer (pH 8). The cDNA library was sequenced on a Nanopore flow cell (FLO-MIN106). Enriched polyA+ RNA was used to construct the library for direct RNA sequencing (SQK-RNA001) according to the manufacturer's instructions. In brief, 400 ng of total polyA+ RNA was used for

reverse transcription adapter (RTA) ligation before subjected to reverse transcription. RNA/DNA hybrids were purified with RNAClean XP beads and ligated with RNA adapters as direct RNA sequencing library. The constructed library was sequenced on the Nanopore flow cell (FLO-MIN106).

**Circularized chromosome conformation capture (4C).** 4C-seq was performed as described previously[37] with modifications. K562 cells were fixed as ChIP protocol except that the cells were fixed in 2% formaldehyde at the concentration of 1 million cells per mL for 10 min. Cells were lysed in complete lysis buffer (10 mM Tris-HCl, pH 8, 10 mM NaCl, 0.2% NP-40, and complete protease inhibitor cocktail with EDTA (Roche)). The nuclei were pelleted, washed once with complete lysis buffer, and then lysed with 100 μl 0.5% SDS solution at 62 °C with 400RPM agitation for <10 min. 299 μl nuclease-free water and 20 μl of 20% Triton X-100 were added before 15 min incubation at 37 °C to quench SDS. 1X DpnII buffer (NEB), 0.8 U/μl DpnII restriction enzyme (NEB), and 0.08 mg/μl BSA were supplemented to the chromatin solution for DpnII digestion at 37 °C for 4 h with 1000 RPM agitation. Additional 400U of DpnII was added to the sample for overnight digestion. DpnII was heat-inactivated at 62 °C for <20 min. An aliquot of the digested chromatin was sampled for DNA purification followed by DNA quantification, agarose gel electrophoresis, and PCR to monitor digestion efficiency. Digested chromatin equivalent to a total 10 μg of DNA was ligated with 660U HC DNA ligase (Invitrogen, 30 U/μl) in 1X ligation buffer (Invitrogen, supplemented with 1% Triton X-100, 0.1 mg/mL BSA) for 8–10 h at 16 °C without shaking. The reaction was incubated at room temperature for 30 min. The chromatin was reverse-crosslinked with 0.5% SDS and 50 μg/mL of Proteinase K (Invitrogen) at 64 °C overnight. RNA in the chromatin was digested with 2 mg/mL RNase A at 37 °C for 1 h. The ligated DNA was extracted with phenol:chloroform:isoamyl alcohol (25:24:1) followed by chloroform and then precipitated with ethanol (68% to avoid SDS precipitation) in the presence of glycogen. The ligated DNA product was analyzed by agarose gel electrophoresis, and the DNA concentration was determined by Qubit DNA HS assay kit.

Three micrograms of the DNA product was further digested with 3U of specific second cutter (NlaIII, MseI, or MluCI). Restriction enzyme was heat-inactivated by incubating the chromatin at 65 °C for 20 min (MluCI was heat-inactivated in the presence of 1.17% SDS). The double-digested DNA was analyzed by agarose gel electrophoresis prior to phenol:chloroform:isoamyl alcohol (25:24:1) extraction and ethanol precipitation in the presence of glycogen. One microgram of purified DNA was ligated with HC DNA ligase (Invitrogen, 30 U/μl) at 20 U/μg DNA in 1X ligation buffer (Invitrogen). For NlaIII or MulCI digested samples, the ligation was performed at 16 °C overnight. For MseI digested samples, the ligation was performed at 4 °C for 24 h. The ligated DNA was recovered by phenol:chloroform:isoamyl alcohol (25:24:1) extraction and ethanol precipitation. 800 ng DNA from each of the biological triplicates were pooled to prepare 4C library (2.4 μg of DNA in total). The library for each viewpoint was constructed by PCR with 0.05 μM 4C adapter primers (see primers section) and 2 ng/μl DNA templates using KAPA HiFi HotStart ReadyMix (KK2602). The libraries were subjected to size selection (250–500 bp) on a 4–20% TBE PAGE gel (Thermo Scientific). The recovered libraries were sequenced on the Illumina Nextseq platform using Nextseq 500 mid-output kit with 2 × 150 paired-end.

**Bioinformatics.** For ChIP-seq analysis, reads were trimmed of adapter sequences by TrimGalore[38] and were mapped by bowtie2[39] against human reference genome GRCh38. The PCR duplicates were removed by SAMtools rmdup[40]. The mapping results were converted into signals by BEDTools[41] in bedGraph format. Then they were converted to bigWig format by bedGraphToBigWig[42,43] for visualization on UCSC genome browser[44]. Peak calling was performed by MACS2[45].

RNA-seq reads were mapped by STAR[46] against human reference genome GRCh38 with GENCODE transcriptome annotation (v26)[47]. RNA-seq reads were trimmed of adapter sequences by TrimGalore[38] and then mapped by STAR to the human reference genome GRCh38 with reference gene annotation GENCODE 26[46,47]. PCR duplicates were removed in the paired-end alignments by SAMtools rmdup. Alignments with mapping quality <20 were removed. Gene expression levels in FPKM were determined by cuffdiff[48]. RNA-seq read counts are generated by featureCounts[49].

For RIP-seq, reads from the RIP data set and its control data set were trimmed of adapter sequences by TrimGalore[38] and were mapped respectively by STAR[46] against the human reference genome GRCh38 with GENCODE[47] transcriptome annotation (v26). The resulting alignments were separated into two parts. (1) The exonic part consisted of alignments belonging to GENCODE annotated transcripts. (2) The non-exonic part consisted of the other alignments. Based on the number of reads that can be mapped into the transcriptome, the larger data set was shrunk down to fit the size of the smaller one. Then the normalized read count of each genomic position was generated for the EZH2 RIP data set and IgG control data set, respectively. The read coverage of each position was defined as the average of normalized read counts within ±150 bp region. Based on the comparison of the read coverage between the EZH2 RIP and the IgG control, sites with ≥2-fold enrichment and Poisson distribution $p$-value ≤10$^{-5}$ were defined as peaks. Each peak was extended to surrounding areas until the fold enrichment dropped below 2. Peaks that can be found in both replicates within 1000 bp range were merged and considered reproducible peaks. Sequences covered by reproducible peaks were extracted for further motif discovery analysis. The upstream

and downstream sequences of reproducible peaks were also extracted and served as the background sequences in the motif finding analysis. Motif discovery is performed with the MEME suite using differential enrichment mode[20,50].

For Nanopore sequencing, raw read collected from MinKNOW were converted to FASTQ file by Albacore 2.3.3 with the following arguments: -k SQK-RNA001 -f FLO -MIN106 -o fastq for direct RNA sequencing; or the following arguments: Albacore 2.3.1 with - -flowcell FLO-MIN106 - -kit SQK-PCS108 - -output_format fastq for cDNA sequencing. For direct RNA sequencing, the reads were mapped to hg38 by Minimap2[51] with -ax splice -uf -k14. The bam files were converted to bed12 by bamToBed -bed 12 and were converted to bigBed file by bedToBigBed -type = bed12. Only uniquely mapped reads were retained. For cDNA sequencing, the reads were mapped to hg38 by LAST with lastal -d90 -m50 -D10 and last-split -d2[52,53]. The maf files were converted to psl file with maf-convert -j1e6 psl. The psl files were subsequently converted to bam file with psl2sam.pl and cigar_tweaker scripts within the SAMtools package[40]. For visualization, the read alignments were converted into bedGraph by BEDTools[42], and finally, into bigWig format by bedGraphToBigWig[42,43].

For 4C-seq analysis, the long-range genomic interaction regions generated by the 4C-Seq experiment were analyzed by the R package r3CSeq v1.30.0[54]. Briefly, for each replicate, the raw reads were aligned to the masked version of the reference human genome (masked for the gap, repetitive and ambiguous sequences) downloaded from the R Bioconductor repository (BSgenome.Hsapiens.UCSC.hg19.masked). The chromosome 11 was selected as the viewpoints, and DpnII was used as the restriction enzyme to digest the genome. A non-overlapping fragment was selected to identify the interaction regions. BAM files were converted to read coverage by bedtools genomecov. The read coverage was normalized according to the sequencing depth. The read coverage difference of the treatment and the control was calculated based on the normalized read coverage. The resulting bedGraph files were converted to bigWig format by bedGraphToBigWig. The genome version of bigWig files was then converted to hg38 by CrossMap[55]. The profile of read coverage difference on genomic intervals was plotted by deepTools[56].

**Statistics and reproducibility**. R was used for statistical analysis. Details about statistical methods and sample numbers used in each experiment were elaborated accordingly within each figure legends.

**Reporting summary**. Further information on research design is available in the Nature Research Reporting Summary linked to this article.

## Data availability
The NGS data generated in this study have been deposited to the NCBI Gene Expression Omnibus (GEO) under accession number GSE141083. Source data of the main figures are available in Supplementary Data 2. Plasmids are available upon reasonable requests.

## Code availability
Softwares and arguments used in this study have been extensively described in the "Methods" section.

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

## Acknowledgements
This work was supported by the National Research Foundation Singapore and the Singapore Ministry of Education under its Research Centres of Excellence initiative, the Singapore Ministry of Health's National Medical Research Council under its Singapore Translational Research (STaR) Investigator Award (MOH-STaR18nov-0002), and the Singapore Ministry of Education Academic Research Fund Tier 3 (MOE2014-T3-1-006 to D.G.T.). D.G.T. is also funded by NIH grant 1R35CA197697 and P01HL131477. L.C. is funded by NIH grant P01HL095489. We also acknowledge Dr. Zheng Zhong for the help in revising the manuscript.

## Author contributions
W.W.T. and X.C. conceived, designed, and conducted experiments; collected and analyzed data; and wrote the manuscript; C.S.W., H.K.T., H.Y., and X.C. performed the data curation and bioinformatics analysis; H.K.T., Q.Z., C.G., K.V., S.S.K., and G.J.M. conducted experiments, analyzed data, and reviewed the manuscript; L.C. and D.G.T. conceived and designed experiments, reviewed and wrote the manuscript.

## Competing interests
The authors declare no competing interests.
