## [Peer Review File · Communications Biology]

Reviewers' comments:

Reviewer #1 (Remarks to the Author):

In this manuscript, Teo and coauthors investigate the role of lncRNA LEVER in the regulation of the globin cluster. In K562 cells, they initially identify LEVER as an EZH2 binding RNA in RIP-seq experiments. Suppression of LEVER expression by CRISPR impacts positively on the neighboring e-globin gene, as does overexpression of EZH2, and this is concomitant with increased levels of H3K27me_{2/3} along the LEVER gene. Finally, 4C analysis of chromatin interactions indicates that LEVER KO cells display reduced interactions with the LCR regulatory region and the e-globin locus, whereas e-globin⁻:LCR interaction is somewhat increased and might cause the positive output in e-globin expression.

Altogether, these data provide some relevant insights into the role of RNAs and lncRNAs in particular as regulators of gene expression through their interaction with Polycomb complexes. This is an area that has been extensively studied, but the added interest of this work lies on the characterization of LEVER RNA near the globin cluster. Although the manuscript is of potential interest for the readership of Communications Biology, there are some key aspects that need further investigation, in particular the involvement of LEVER RNA per se in the regulation.

Major points:

1. The authors convincingly show that disruption of LEVER transcription by means of CRISPR results in an increase in K27me₃ levels at LEVER locus and an increase in e-globin expression. However, this might not necessarily imply a function for LEVER RNA, and changes in local chromatin as a result of suppressed transcription might account for the impact on the neighboring e-globin gene. The only data pointing to a RNA role in the regulation is shown in Figure 3C with the overexpression of LEVER fragments. To support the authors claims that the RNA transcript has a central role per se, LEVER RNA should be directly targeted (e.g., by means of LNA-Gapmers or similar agents), and the levels of H3K27me₃ on LEVER locus and e-globin expression measured. In the absence of this additional data, the title "non-coding RNA LEVER mediates..." or the caption on Fig. 3 "PRC2 mediates the repressive effect of LEVER RNA on e-globin" should be tempered down.
2. In Fig. 3A-B, to link EZH2 function with globin regulation through the impact on LEVER gene, the authors need to show how K27me₃ levels at the LEVER locus change upon the different conditions of EZH2 KO/OE.

Minor points:

1. The choice of K562 cell line needs to be better justified in the text.
2. Pg 15, lin 11-12: "...RNase A treated K562 cells showed a global increase in H3K27me₃ but not total H3 in the input samples...". This is not so clear from western blot in Suppl. Fig. 2C, since total H3 seems also increased in the RNase A condition. Please provide quantitation of the signal.
3. Fig. 2F, a-tubulin is overexposed and thus not useful as loading control. Please provide a less exposed image.
4. Suppl. Figure 3E, could the authors broaden the analysis to include a panel of cell lines where e-globin expression is detectable?
5. Fig. 3B, the levels of e-globin in K562 (WT) cells are hard to estimate. Please provide a clearer WB for this experiment.
6. Pg 25, lines 5-6: "It should be noted that this PRC2-inhibiting effect was achieved in a global, rather than a localized manner". What does this mean?
7. Fig. 3C, a better description of LVR-1/2 MeMe fragments is needed, maybe with a diagram of their location on the transcript. What is their length? A control fragment from LEVER RNA of similar length but not bound

Reviewer #2 (Remarks to the Author):

Tenen and colleagues report that a lncRNA they call LEVER negatively regulates e-globin expression. They begin with fRIPseq experiments to find RNAs bound to the EZH2 subunit of PRC2 in human K562 cells. They find that transcriptional inhibition or RNase treatment cause increased binding of PRC2 to chromatin. They then move to one of the PRC2-bound RNAs, the LEVER

lncRNA, which is transcribed 236 kb from the globin locus, and show that LEVER knock-out or dCas9 knock-down both increase e-globin expression at the RNA and protein level. This is the opposite effect that one might expect from their genome-wide data, because knock-down of a PRC2 inhibitor might be expected to increase PRC2 binding and repress the chromatin. They then move to 4Cseq experiments to look at nuclear architecture, from which they argue that derepression of the LEVER chromatin by LEVER lncRNA causes a change in nuclear organization that represses e-globin transcription.

This is certainly a fascinating system, and the experiments have good controls and are generally well done. Although one can always think of more experiments that would enhance the story, in this case I think that the experiments provided are complete enough to warrant publication. There are, however, a number of points that should be clarified, as follows.

1. What's needed in Fig 1B is a standard fRIPseq plot of pull-down/input (x-axis) vs P-value or some other statistical measure (Y-axis), and the current plots in 1B can be moved to Supplemental. The fRIPseq plot should have the dot representing LEVER clearly indicated.
2. The two plots currently in Fig 1B should both have the LEVER RNA clearly marked.
3. P 16 line 13, please add explanation for why you used poly(A)-plus RNA when you found that most hits were poly(A)-minus.
4. P 25 lines 3-5, it would be good to come back to this in the Discussion; LEVER can act in trans, but does it normally act in cis or in trans or can't you tell?
5. P 27, legend to A, if this is one biological sample, then what do error bars represent?
6. Fig 4C and Supple Fig 5A show H3K27ac, so this should be mentioned in text.
7. The Discussion would benefit from more careful statements about what is actually shown and what is proposed as a reasonable model from the data. As one example, p. 41 line 22 states as a fact "that further regulate," but the data show correlation not causation. I suggest a paragraph in the Discussion describing "loose ends" that the paper has not tied up, and perhaps future experiments that could further test the model presented here.

We thank the reviewers for their suggestions. We attempted to directly address all of their comments, performing additional experiments and bioinformatic analyses, and revised the manuscript to address the reviewers' questions. In particular, we analyzed the H3K27me3 levels at the LEVER locus upon EZH2 knockout/overexpression and generated new data with supported our proposed mechanism. We also performed the suggested RIP-seq analysis and revised Fig.1b. Specific responses are described below following each comment. We hope the revised manuscript is now suitable for publication in Communications Biology.

~~~~~

## **Reviewers' comments:**

### **Reviewer #1 (Remarks to the Author):**

In this manuscript, Teo and coauthors investigate the role of lncRNA LEVER in the regulation of the globin cluster. In K562 cells, they initially identify LEVER as an EZH2 binding RNA in RIP-seq experiments. Suppression of LEVER expression by CRISPR impacts positively on the neighboring  $\epsilon$ -globin gene, as does overexpression of EZH2, and this is concomitant with increased levels of H3K27me2/3 along the LEVER gene. Finally, 4C analysis of chromatin interactions indicates that LEVER KO cells display reduced interactions with the LCR regulatory region and the  $\epsilon$ -globin locus, whereas  $\epsilon$ -globin:LCR interaction is somewhat increased and might cause the positive output in  $\epsilon$ -globin expression.

Altogether, these data provide some relevant insights into the role of RNAs and lncRNAs in particular as regulators of gene expression through their interaction with Polycomb complexes. This is an area that has been extensively studied, but the added interest of this work lies on the characterization of LEVER RNA near the globin cluster. Although the manuscript is of potential interest for the readership of Communications Biology, there are some key aspects that need further investigation, in particular the involvement of LEVER RNA per se in the regulation.

### **Major points:**

#### **#1-1**

The authors convincingly show that disruption of LEVER transcription by means of CRISPR results in an increase in K27me3 levels at LEVER locus and an increase in  $\epsilon$ -globin expression. However, this might not necessarily imply a function for LEVER RNA, and changes in local chromatin as a result of suppressed transcription might account for the impact on the neighboring  $\epsilon$ -globin gene. The only data pointing to a RNA role in the regulation is shown in Figure 3C with the overexpression of LEVER fragments. To support the authors claims that the RNA transcript has a central role per se, LEVER RNA should be directly targeted (e.g., by means of LNA-Gapmers or similar agents), and the levels of H3K27me3 on LEVER locus and  $\epsilon$ -globin expression measured. In the absence of this additional data, the title "non-coding RNA LEVER

mediates...” or the caption on Fig. 3 “PRC2 mediates the repressive effect of LEVER RNA on  $\epsilon$ -globin” should be tempered down.

### Response:

We appreciate the reviewer for this suggestion. We agree that loss of local transcription may also contribute to the chromatin changes observed in our CRISPR experiments. To study the function of LEVER RNA without interrupting its transcription, we have attempted to use shRNAs as well as the Cas13a RNA targeting system1 to directly knock down LEVER nascent RNA, with multiple shRNA/sgRNA sequences tested for each method. However, we failed to decrease LEVER nascent RNA abundance using both methods (Rebuttal Fig.1). We think this could be due to the natural difficulty to target such a long nuclear nascent transcript of over 200,000 nt.

Rebuttal Fig. 1 LEVER knock-down by shRNA and Cas13a

We (Supplementary Fig.2C) and others2 have validated the PRC2-inhibiting function of RNAs by RNA degradation methods, which is independent of transcriptional processes. Nevertheless, in the absence of direct experimental evidence, we have modified the original title to “**Non-coding RNA LEVER sequestration of PRC2 can mediate long range gene regulation**”, and have modified the caption of Fig.3 (page 21) to “*PRC2 enzymatic function maintains  $\epsilon$ -globin expression in K562*” in the current manuscript. In addition, we added this limitation to the Discussion (page 34, line 23 - page 35, line 7):

*“Considering the reported effects of epigenetic processes in regulating transcription3, the epigenetic and  $\epsilon$ -globin expression changes observed in our CRISPR-based LEVER knock-out/knock-down systems could be caused in part by silencing LEVER transcription event. Due to the difficulty to knock down a nuclear-localized, very long (> 200 kb) nascent transcript without disrupting its transcription through currently available RNA targeting systems, we tried but failed to reduce LEVER nascent RNA abundance*

using shRNA and Cas13a 1. Therefore, the independent effects of LEVER RNA and transcription remain to be resolved in future studies.”

**#1-2.**

In Fig.3A-B, to link EZH2 function with globin regulation through the impact on LEVER gene, the authors need to show how K27me3 levels at the LEVER locus change upon the different conditions of EZH2 KO/OE.

**Response:**

We appreciate the suggestion from the reviewer. We have tested the K27me3 changes at the LEVER locus upon the knock-out and re-expression of EZH2 and its mutants, and we observed a positive correlation between the expression level of  $\epsilon$ -globin (Fig.3a-b) and the H3K27me3 levels within the LEVER locus as was shown in the H3K27me3 ChIP-qPCR results at the LVR-4, LVR-3, and LVR-2 regions (Rebuttal Fig.2). To accommodate the logical flow of the manuscript, we have added this figure as Fig.4d (page 25) and also added the following description on page 22 line 24 to page 23 line 3:

“Besides, we observed a positive correlation between  $\epsilon$ -globin expression (Fig.3a, 3b) and H3K27me3 levels within the LEVER locus (Fig.4d) in EZH2 knocked-out and rescued cells, which suggests the importance of PRC2-mediated epigenetics within the LEVER locus in regulating  $\epsilon$ -globin expression.”

Rebuttal Fig.2 H3K27me3 ChIP in EZH2 KO/OE Cells

Minor points:  
#1-3.

The choice of K562 cell line needs to be better justified in the text.

**Response:**

The initial aim of our study is to investigate the RNA binding property of PRC2 and its biological functions. We chose to use the K562 cell line for RIP-seq due to its common use in genomic/epigenomic studies (e.g., there is a large amount of NGS data available from both public databases and our lab), and from the RIP-seq we identified LEVER, which was found to regulate globin gene expression in this cell line. We agree that K562 is not the best system to study regulation of globin genes. However, as we were most interested in the molecular mechanism of how PRC2-RNA interactions regulate neighboring gene expression, we continued with K562 considering 1) the relative ease of conducting genetic modifications to achieve LEVER and EZH2 knock-out/knock-down in this cell line; and 2) the practicability to obtain sufficient starting materials for mechanistic studies (eg., ChIP and 4C), which require large number of cells to achieve reproducible results. We have discussed the above reasons for using K562 as well as its limitations in the added paragraph in the discussion section on page 35 line 15-22 as:

*“Finally, we used K562 cell line as our model system to study the molecular mechanism of PRC2-LEVER interaction in regulating  $\epsilon$ -globin expression. The cell line was chosen due to its feasibility for genetic modifications and its culturability to obtain sufficient materials for ChIP and 4C assays that require large numbers of cells for reliable results. However, the exact physiological importance of LEVER in regulating  $\epsilon$ -globin expression will require future studies in more physiologically relevant systems, eg., in human hematopoietic stem and progenitor cells.”*

**#1-4.**

Pg 15, lin 11-12: “...RNase A treated K562 cells showed a global increase in H3K27me3 but not total H3 in the input samples...”. This is not so clear from western blot in Suppl. Fig.2C, since total H3 seems also increased in the RNase A condition. Please provide quantitation of the signal.

**Response:**

We appreciate the reviewer for pointing this out. We quantified the band density using ImageJ (Rebuttal Fig.3). As the reviewer pointed out, the total H3 signal of the RNase A-treated sample is 1.4-fold over the Mock sample. Yet the H3K27me3 signal of the RNase A-treated sample is 5.1-fold over the Mock sample, so it is 3.6 fold after normalization. To avoid confusion, we changed the sentence on page 11 line 11-12 to read:

“...RNase A treated K562 cells showed a global increase in H3K27me3 relative to total H3 in the input samples...”

Rebuttal Fig.3 Quantification of Supplementary Fig.2c (revised Supplementary Fig.2d)

**#1-5.**

Fig. 2F,  $\alpha$ -tubulin is overexposed and thus not useful as loading control. Please provide a less exposed image.

**Response:**

We have changed Fig.2f with a less exposed image for the  $\alpha$ -tubulin loading control in our revised manuscript (Rebuttal Fig.4). Besides, we also attach here the original film scan of this blot (Rebuttal Fig.5). As suggested by the equal amounts of non-specific bands that are not overexposed, we believe the protein loading is even between the four samples.

Rebuttal Fig.4

Rebuttal Fig.5 Original Film Scan for Fig.2f α-tubulin loading control

**#1-6.**

Suppl. Figure 3E, could the authors broaden the analysis to include a panel of cell lines where ε-globin expression is detectable?

**Response:**

We analyzed the RNA-seq data from the *leukemia and lymphoma cell line panel* (LL-1004). From these cell lines, only four showed detectable ε-globin expression, and we observed a reverse correlative trend between ε-globin and LEVER in these cells (Rebuttal Fig.6).

Rebuttal Fig.6 RNA-seq analysis of LEVER and ε-globin in four blood cancer cell lines

We further examined LEVER expression in two erythroid cell lines available in our lab (HEL and TF-1) by RT-qPCR (Rebuttal Fig.7). The reverse correlation between ε-globin and LEVER expression was further supported.

Rebuttal Fig.7 RT-qPCR analysis of LEVER and ε-globin in two blood cancer cell lines

We have added this data as supplemental figure 3f and 3g in the current manuscript, with following description added on page 18 line 4-8:

*“...as well as in a panel of leukemia and lymphoma cell lines that express ε-globin (Supplementary Fig.3f) 4. A similar anti-correlation between LEVER and ε-globin was manifested as in the K562 cells, and this correlation was further supported by RT-qPCR of HEL and TF-1 cell lines (Supplementary Fig.3g).”*

#1-7.

Fig. 3B, the levels of  $\epsilon$ -globin in K562 (WT) cells are hard to estimate. Please provide a clearer WB for this experiment.

**Response:**

We have replaced the  $\epsilon$ -globin and the corresponding  $\beta$ -globin loading control blots in our revised manuscript on page 20 (Rebuttal Fig.8).

Rebuttal Fig.8

#1-8.

Pg 25, lines 5-6: “It should be noted that this PRC2-inhibiting effect was achieved in a global, rather than a localized manner”. What does this mean?

**Response:**

We apologize for the confusion caused by this sentence. For the LVR-1/2 MeMe overexpression assay, we would expect the abundance of LEVER fragments to be increased throughout the nucleus (globally), instead of being restricted to the LEVER locus. **To avoid being misleading, we have removed this sentence from the current manuscript.**

#1-9.

Fig. 3C, a better description of LVR-1/2 MeMe fragments is needed, maybe with a diagram of their location on the transcript. What is their length? A control fragment from LEVER RNA of similar length but not bound

**Response:**

We apologize for not giving a proper description for the LVR-1/2 MeMe fragments, and we have appended a diagram as Supplementary Fig.4a on **Supplementary file** page 9 (Rebuttal Fig.9).

In short, we selected ~300nt G/A rich fragments (blue box with sequence listed below) from the RIP enriched peaks of LVR-1 and LVR-2 (red bars with arrows). As the G/A preference of PRC2 has been validated by previous publications from different groups 5,6, we hypothesized that these selected fragments with the validated motif should have stronger PRC2 binding affinity than other regions within the whole LEVER transcript.

While a non-PRC2-binding fragment from the LEVER transcript would theoretically be a proper negative control for this experiment, it is difficult to locate such a fragment on LEVER from our RIP-seq analysis. Our RIP-seq analysis set stringent peak-calling thresholds to reduce false-positive candidates, and this is at the expense of losing potential hits. In other words, we cannot rule out the possibility that a region not identified as peaks by our analysis may still bind to EZH2 with modest affinity. Considering the possibility of choosing a false-negative control fragment from LEVER, alternatively, we used a non-translatable EGFP RNA fragment (~700nt), as we believe this exogenous non-G/A-rich fragment should serve as a true negative control.

To elaborate the above descriptions, we also added the following sentences to our current manuscript on page 19 line 8-14:

*“In short, we selected ~300nt G/A rich fragments from the RIP enriched peaks of LVR-1 and LVR-2. As the G/A preference of PRC2 has been validated by previous publications from different groups 5,6, we hypothesized that these selected fragments with the validated motif should have stronger PRC2 binding affinity than other regions within the whole LEVER transcript. Meanwhile, we used a non-translatable non-G/A-rich EGFP RNA fragment (~700nt) as a negative control.”*

```
>chr11:5337598-5337847 (250nt)
CCTGTAATCTCAGCTACTCAGGAGGCTGAGGCAGGAGAAATTGCTTGAACC
CAGGAGGCAGAGGTTGCAGTGAGCTGAGATCGCACCCACTGCACCTCTAGCA
TGGGTGAAAAGAGTGAGACTTCATCTCAAATAAAGAAAAATACAATTAAA
TACAGAGCTTCTAAACAGCAAAATAAACTATCATCAGAGCGAACAGGCAA
CCTACAGAATGGGAGAAAAATTTCTGCAATCTACCCATCTGACAAAGGTCT
```

```
>chr11:5401788-5402156 (369nt)
GCCTGTAGTCCCAGCTACTTGAGAGGCTGAGGTGGGAAGATCACTTGGGC
CCAGAAGGCAAGGTTGCAGTGAGCCAGGATTGCACGCCTACACTTATGC
CTGGTAACAGCATGAGACAAAAGAAAAGGAGAGAAAAGAAAAGAAAG
AAAAAGAGAACAGCAAGCGGGAGGGAGGGAAGGAGAAAAGGAAGGAA
AGAAGAGAGGAAAGAAATAAAGAAAAGGAAAGAGAGAGAGAGAGAA
AGAAAAGGAAGGAAGGAGAGAGAAAAGGAAAGGAAAAGAAAAGAAA
GCAGAGAAAAGAGGGAGGAAGGAATGATAGAAAAGAAAAGAAAAGAA
AGATTATGCATTTAAGGTG
```

Rebuttal Fig.9 Annotation of LVR-1/2 MeMe

## Reviewer #2 (Remarks to the Author):

Tenen and colleagues report that a lncRNA they call LEVER negatively regulates  $\epsilon$ -globin expression. They begin with fRIPseq experiments to find RNAs bound to the EZH2 subunit of PRC2 in human K562 cells. They find that transcriptional inhibition or RNase treatment cause increased binding of PRC2 to chromatin. They then move to one of the PRC2-bound RNAs, the LEVER lncRNA, which is transcribed 236 kb from the globin locus, and show that LEVER knock-out or dCas9 knock-down both increase  $\epsilon$ -globin expression at the RNA and protein level. This is the opposite effect that one might expect from their genome-wide data, because knock-down of a PRC2 inhibitor might be expected to increase PRC2 binding and repress the chromatin. They then move to 4Cseq experiments to look at nuclear architecture, from which they argue that derepression of the LEVER chromatin by LEVER lncRNA causes a change in nuclear organization that represses  $\epsilon$ -globin transcription.

This is certainly a fascinating system, and the experiments have good controls and are generally well done. Although one can always think of more experiments that would enhance the story, in this case I think that the experiments provided are complete enough to warrant publication. There are, however, a number of points that should be clarified, as follows.

**#2-1.**

What's needed in Fig 1B is a standard fRIPseq plot of pull-down/input (x-axis) vs P-value or some other statistical measure (Y-axis), and the current plots in 1B can be moved to Supplemental. The fRIPseq plot should have the dot representing LEVER clearly indicated.

**Response:**

We appreciate this suggestion from our reviewer.

To our knowledge, the standard fRIP-seq analysis technically resembles RNA-seq analysis. In other words, in the standard fRIP-seq analysis, fold enrichment per transcript/gene is calculated using standard differential expression analysis algorithms (eg. Cuffdiff, as was used by Hendrickson et. al 7) for all the annotated transcript/genes. Then, empirical fold change and/or p-value thresholds are chosen to identify statistically significant RNAs that the immunoprecipitated protein binds.

However, these standard fRIP-seq analyses rely on the existing transcript annotation. So only **known gene loci** and **annotated spliced transcripts** are considered in the analysis. Moreover, these standard analyses **treat each annotated spliced transcript as a unit for read-counting and comparisons**. We think this methodology does not fit our purpose for the following reasons:

1) Considering the promiscuous RNA-binding nature of EZH2, besides annotated genes, we are also interested in intergenic transcripts that are not included in the annotation and are therefore ignored by the standard fRIP-seq analyses.

2) We are interested in the nascent transcriptome that includes intron regions. If we want to use the standard fRIP-seq analysis pipelines to do this, we have to substantially revise the publicly available transcriptome annotation as was described by Hendrickson et al 7.

3) Considering the potential EZH2 binding heterogeneity within a transcript, we hoped to use a method that can narrow down EZH2 binding regions within a transcript. However, the standard fRIP-seq analysis pipelines generate transcript/gene-level output (i.e., one read count value per transcript/gene), which does not allow us to study the EZH2 binding heterogeneity within a transcript.

To realize the above purposes, we developed a pipeline (unpublished, manuscript in preparation) that more resembles ChIP-seq rather than RNA-seq analyses. Specifically, our ChIP-seq-like methods consider the reads mapped to the whole genome instead of only the transcriptome, and the read counts are compared per genomic fragment rather than per transcript/gene. Significant binding regions are identified as peaks with ChIP-seq peak-calling algorithm MACS2 by comparing normalized IP and IgG samples. As we wanted to identify strong hits to follow up, we set stringent cut-offs for both fold enrichment ( $\geq 1.7$ ) and Poisson distribution p-value ( $\leq 10^{-3}$ ) at the peak-calling step that identified a short list of 1528 reproducible peaks annotated to only 299 gene loci. We found that EZH2 interacts substantially with intron and intergenic regions (Fig. 1A), which proved the importance to include these regions in the analyses for EZH2 RIP. Besides, these 1528 peaks successfully enriched the known EZH2-binding G/A rich motif (Supplementary Fig. 2b).

As the reviewer requested, we plotted the pull-down/**IgG** fold enrichment vs p-value for these RIP peaks (Rebuttal Fig. 10). We noticed that **12 peaks were mapped to the LEVER transcript**, highlighting LEVER as a genuine EZH2-interacting RNA in K562. But notably, compared to the plot of pull-down/input fold change vs P-value in the standard fRIPseq analysis, where each dot represents one transcript and all transcripts (no matter significant or not) are shown, our plot **only shows those significant peaks** identified by our peak-calling algorithm (Rebuttal Fig. 10).

Rebuttal Fig.10 Fold enrichment –  $p$ -value plot of EZH2 RIP-seq identified peaks. One dot represents one individual peak. The 12 peaks annotated to LEVER transcript are labeled as red dots.

Due to the differences between the standard fRIP and our RIP-seq analyses, we think this figure (Rebuttal Fig.11) may confuse the readers, as only those strongest hits are analyzed and shown. So we prefer to add this figure as a supplementary figure (Supplementary Fig.2a) on **Supplementary file** page 3 in the revised manuscript.

#2-2.

The two plots currently in Fig 1B should both have the LEVER RNA clearly marked.

### Response:

We have revised Fig.1B and have labelled the three **overlapped RIP peaks** (we further combined the 12 identified peaks as stated in #2-1 into three) from LEVER locus (Rebuttal Fig.11) and added description on page 12 line 23 to page 13 line 1 as:

*“The nascent origin of these PRC2-interacting LEVER fragments were further supported by their enrichment in the nuclear and polyA- fraction as suggested by the fractionated RNA-seq analyses (Fig.1b)...”*

Rebuttal Fig.12

We noted that LVR-1 showed comparable counts in the poly(A)-plus and poly(A)-minus fractions. Considering that it was not identified as a spliced exon in our poly(A)-enriched Nanopore long-read sequencing (Fig.1d), we do not consider this region to be poly(A)-plus *per se*. Instead, we hypothesize that the higher read-count of LVR-1 in the poly(A) enriched short-read RNA-seq to be caused by its genomic location close to the 3' end of the transcript (Fig.1d).

### #2-3.

P 16 line 13, please add explanation for why you used poly(A)-plus RNA when you found that most hits were poly(A)-minus.

### Response:

The nascent LEVER RNA is ~200,000 nt and has not been properly annotated in the genome. Therefore, we wanted to know whether this long nascent transcript will be processed, i.e., spliced and polyadenylated. The Nanopore sequencing of the poly(A)-plus RNA fraction allowed us to examine whether such a long LEVER locus is transcribed from start to end into a single transcript and if it is spliced. Indeed, through the Nanopore sequencing, we found LEVER RNA is spliced and there are several **exons**. Importantly, comparing these exons with our RIP-seq identified peaks we further knew that the EZH2 interacting parts are the LEVER introns. These results highlight that EZH2 interacts with the nascent transcript of LEVER.

We are sorry for the confusion caused and we have clarified the purpose part in the revised manuscript on page 12 line 14-17 as :

*“To examine if LEVER transcripts undergo splicing and to identify LEVER exons, if any, we performed long-read Nanopore direct sequencing of K562 total polyA+ RNA (Fig.1d) as well as Nanopore cDNA sequencing of K562 nuclear RNA (Supplementary Fig.2e).”*

#### #2-4.

P 25 lines 3-5, it would be good to come back to this in the Discussion; LEVER can act in trans, but does it normally act in cis or in trans or can't you tell?

#### Response:

We agree this was not clear, and agree with the reviewer that these LEVER fragments worked *in trans* in the LEVER MeMe fragment overexpression experiment. The current study focused on the *in cis* function of LEVER, based on the well-accepted assumption that the nascent transcript is tethered to its own locus8,9. But as the reviewer mentioned, we could not rule out the possibility that LEVER RNA, either the nascent transcript or the spliced mature forms, can also act in trans. We have added to the discussion about this in the revised manuscript on page 35 line 11-15 as:

*“While our investigation and proposed model focus on the cis-regulating role of the LEVER nascent transcript based on the assumption that the nascent transcript is tethered to its own locus8,9, we can not rule out the possibility that LEVER RNA, either the nascent transcript or the spliced mature forms, can also act in trans.”*

#### #2-5.

P 27, legend to A, if this is one biological sample, then what do error bars represent?

#### Response:

The error bars here represent technical replicates from the single biological sample. We apologize for not stating this clearly. We have revised all related figure legends to clarify.

#### #2-6.

Fig 4C and Supple Fig 5A show H3K27ac, so this should be mentioned in text.

#### Response:

We appreciate the reviewer for pointing this out. We have added one sentence

*“In addition, H3K27 acetylation (H3K27ac) at the LEVER locus was correspondingly reduced, further supporting the above observations.”*

in our manuscript on page 22 line 18-19 to describe the H3K27ac ChIP-seq results.

## #2-7.

The Discussion would benefit from more careful statements about what is actually shown and what is proposed as a reasonable model from the data. As one example, p. 41 line 22 states as a fact “that further regulate,” but the data show correlation not causation. I suggest a paragraph in the Discussion describing “loose ends” that the paper has not tied up, and perhaps future experiments that could further test the model presented here.

### **Response:**

We apologize for the misleading statements and have revised the sentence to

*“In our work, we reported that the autonomous PRC2 inhibition mediated by nascent RNAs can lead to changes in chromatin interactions and expression of neighboring genes.”*

in our current manuscript on page 31 line 20-22.

In addition, we also added a paragraph (page 34 line 23 to page 35 line 22) to discuss the limitations as well as future experiments of this study in the Discussion section.

### **Other changes made in the revised manuscript:**

1. We have changed the data plotted in Fig.2e (page 14). While we used data measured by the main primer set in most other  $\epsilon$ -globin qPCR figures, we erroneously plotted the data measured by an alternative set in the original Fig.2e. Although we do not doubt the validity of the old figure, we replaced it with the one measured with the main primer to keep the consistency of all  $\epsilon$ -globin qPCR data.
2. All individual data points were labelled on each bar-plot figures.

### **References**

- 1 Abudayyeh, O. O. *et al.* RNA targeting with CRISPR-Cas13. *Nature* **550**, 280-284, doi:10.1038/nature24049 (2017).

- 2 Beltran, M. *et al.* The interaction of PRC2 with RNA or chromatin is mutually antagonistic. *Genome research* **26**, 896-907, doi:10.1101/gr.197632.115 (2016).
- 3 Yuan, W. *et al.* H3K36 methylation antagonizes PRC2-mediated H3K27 methylation. *J Biol Chem* **286**, 7983-7989, doi:10.1074/jbc.M110.194027 (2011).
- 4 Quentmeier, H. *et al.* The LL-100 panel: 100 cell lines for blood cancer studies. *Sci Rep* **9**, 8218, doi:10.1038/s41598-019-44491-x (2019).
- 5 Wang, X. *et al.* Targeting of Polycomb Repressive Complex 2 to RNA by Short Repeats of Consecutive Guanines. *Mol Cell* **65**, 1056-1067 e1055, doi:10.1016/j.molcel.2017.02.003 (2017).
- 6 Beltran, M. *et al.* G-tract RNA removes Polycomb repressive complex 2 from genes. *Nat Struct Mol Biol* **26**, 899-909, doi:10.1038/s41594-019-0293-z (2019).
- 7 D, G. H., Kelley, D. R., Tenen, D., Bernstein, B. & Rinn, J. L. Widespread RNA binding by chromatin-associated proteins. *Genome Biol* **17**, 28, doi:10.1186/s13059-016-0878-3 (2016).
- 8 Gil, N. & Ulitsky, I. Regulation of gene expression by cis-acting long non-coding RNAs. *Nat Rev Genet* **21**, 102-117, doi:10.1038/s41576-019-0184-5 (2020).
- 9 Werner, M. S. & Ruthenburg, A. J. Nuclear Fractionation Reveals Thousands of Chromatin-Tethered Noncoding RNAs Adjacent to Active Genes. *Cell Rep* **12**, 1089-1098, doi:10.1016/j.celrep.2015.07.033 (2015).